# Study on characterization of sandstone pore structure and seepage mechanism based on NMR and CT technology

**Songxin Zhao, Kaide Liu** ⓘ*, **Chaowei Sun, Wenping Yue, Qiyu Wang, Yu Xia, Xinping Wang**

Shaanxi Key Laboratory of Safety and Durability of Concrete Structures, Xijing University, Xi'an, China

* liukaide2006@163.com

## Abstract

Coal-seam roof water disasters severely constrain safe and high-efficiency coal production. Integrating NMR, X-CT, Avizo, Netfabb, and Comsol, we systematically characterized pore architecture and flow behavior for the coarse, medium, and fine sandstones of the Luohe Formation in the Gaojiapu coal mine, Huanglong Jurassic coalfield, Ordos Basin. Results show that the NMR cumulative porosity of coarse, medium, and fine sandstones is 14.36%, 17.82%, and 16.09%, respectively; NMR permeability is 38.77 mD, 1.46 mD, and 0.87 mD; the proportions of macropores and micro-fractures are 48% and 27%, 34% and 12%, and 46% and 8%; fully movable porosity accounts for 68%, 42%, and 29%; NMR-connected pore fractal dimensions are 2.847, 2.943, and 2.955; CT total porosity is 16.43%, 15.51%, and 14.68%; CT connected porosity is 16.19%, 15.21%, and 14.36%; connectivity is 98.54%, 98.07%, and 97.82%; mean pore diameter is 17.50 μm, 17.16 μm, and 17.03 μm; mean throat radius is 32.63 μm, 29.11 μm, and 28.75 μm; absolute permeability is 10.88 D, 10.56 D, and 8.40 D; tortuosity is 1.63, 1.65, and 1.69. Flow-field simulations indicate that velocities below 0.08 m/s dominate, covering 80.14% in coarse sandstone and 97.54% in fine sandstone, providing a theoretical basis for roof water-hazard prevention.

## 1. Introduction

Roof–water inrush has long been a critical geological barrier to the safe and high–efficiency extraction of coal resources [1,2]. The Huanglong Jurassic coalfield in the Ordos Basin, one of China's major energy bases, is threatened by the huge Luohe Formation sandstone aquifer that directly overlies the workable seams [3,4]. To tackle this problem, previous studies have systematically documented lithofacies characteristics, rock mechanical properties, and the height of the "two–zone" (caving + fractured) development [4–15]. It has been demonstrated that the primary

**Data availability statement:** All relevant data are within the manuscript and its Supporting Information files.

**Funding:** This study was supporting by the National Nature Science Foundation of China: (No.52104222 to WY and No. 51909224 to CS), the Natural Science Foundation Research Project of Shaanxi Province: (2019JM-182 & 2021JLM-48 to KL and 2025JCBMS-511 to CS) and the Special Fund for High-level Talents of Xijing University: (XJ18T04 to KL and XJ24B12 to CS).

**Competing interests:** The authors have declared that no competing interests exist.

fissures embedded in the water–bearing sandstone provide the essential skeleton for the initiation and propagation of water-conducting fracture networks [11], while the micro-porous architecture of the sandstone itself offers the necessary storage space and flow pathways for confined groundwater [6,16–18]. Therefore, systematically and comprehensively understanding the characteristics of the microscopic pore structure and the seepage mechanism of the thick water-bearing Luhe Formation sandstone is also a key part of accurately formulating the prevention and control plan for the roof sandstone water hazard, and it is of great significance for effectively preventing and controlling the roof sandstone water hazard.

Extensive studies worldwide have focused on the micro-pore architecture and flow mechanisms of sandstones. Regarding pore-structure characterization, Wang et al. [19] combined NMR with fractal theory and demonstrated that freeze–thaw cycles enlarge macropores and throats, leading to a linear increase in permeability and a reduction in fractal dimension. This identifies macropores as the most freeze-sensitive feature. Mu et al. [20] introduced a trimodal NMR-porosity index and a throat-radius index to quantify the complex pore network of tight Chang-8 and Chang-9 sandstones in the Zhijing–Ansai area of the Ordos Basin, markedly improving the accuracy of log interpretation. Mondal et al. [21] employed NMR to characterize the pore structure of complex Eocene carbonates from an offshore oilfield in India. By correlating NMR $T_2$ distributions with capillary-pressure data, they derived a conversion coefficient, performed radius-dependent porosity partitioning, and conducted a two-stage fractal analysis, revealing a clear relationship between fractal dimension and petrophysical parameters. Zhang et al. [22] used mercury-injection capillary pressure (MIP) and NMR to analyze the pore structure of coal-measure sedimentary rocks (shale, mudstone, and sandstone) and determined pore-size distributions and fractal dimensions. Zhao et al. [23] utilized X-ray computed tomography (CT) to realize three-dimensional visualization and refined quantitative characterization of rock micro-structures. Zheng et al. [24] combined micro-CT digital-core technology with Avizo-based algorithms to construct digital-core models, enabling three-dimensional visualization and quantitative analysis of the micro-pore structure of tight sandstone samples from the Stone-Box Formation in the northern J area of the Ordos Basin, thereby providing technical support for quantitative evaluation of reservoir micro-parameters and for visualization and quantitative characterization of sample micro-structures.

In terms of flow mechanisms, Wang et al. [25] employed CT scanning to investigate the pore structure and mechanical properties of porous rock in mining strata, revealing spatial consistency between the pore network model (PNM) and crack propagation paths; by integrating a network model with pore topology, they enabled pore-scale simulation and prediction of permeability. Wang et al. [26] acquired sandstone pore-structure data via CT and performed multiscale flow simulations with network models to clarify how connectivity, throat-size distribution, and grain arrangement control fluid movement. Zhao et al. [27] combined CT with pore-network modeling to quantify the impact of topological parameters such as connectivity and branch number on permeability in sandstone reservoirs and proposed an improved

pore-network simulation method. Liu et al. [28] extracted pore structures from CT images, predicted flow capacity with network models, examined the influence of connectivity and branch number on permeability, and presented an upgraded network approach. Zhang et al. [29] constructed pore-network models from CT data for high-porosity sandstones, analyzed fluid flow, studied the effects of connectivity and throat-size distribution on permeability, and developed an improved network algorithm. Liu et al. [30] built equivalent pore-network models from micro-CT images using image-processing algorithms, realized pore-scale flow simulations, and computed absolute permeability, offering new insights into sandstone flow behavior. Liu et al. [31] reconstructed micro-pore geometries directly from image files with COMSOL and conducted pore-scale flow simulations, validating the feasibility of image-based geometric reconstruction and providing a novel method for investigating micro-flow mechanisms in low-permeability reservoirs.

Although abundant studies have addressed sandstone pore structure and flow behavior, the two aspects are usually treated separately, and comparative analyses of different sandstone types within a single stratigraphic unit remain scarce and fragmentary. To fill this gap, we present an integrated investigation of coarse, medium, and fine sandstones widely occurring in the Luohe Formation of the Gaojiapu coal mine, Huanglong Jurassic coalfield, Ordos Basin. NMR and X-CT experiments were combined with Avizo, Netfabb, and COMSOL workflows to characterize pore architecture and pore-scale flow, aiming to provide a theoretical basis for roof-water-inrush prevention. The Luohe sandstone aquifer is exceptionally thick (average 400.35 m) and highly water-rich (maximum specific capacity 2.288 L s$^{-1}$ m$^{-1}$). The parting between the aquifer and the coal seam is only ~84 m, giving rise to a measured water pressure of 7.4 MPa. The upper-middle and lower-middle sub-sections are the main water-bearing intervals, yet they are neither drainable nor easily groutable because of low fracture intensity, limited grout take, and a diffusion radius of only a few meters, posing a serious challenge to water-pressure reduction by grouting.

## 2. Experimental methods

### 2.1. NMR experiments

The detailed NMR experimental steps are shown in Fig 1, specifically as follows: ① Sample origin: Core samples of Luhe Formation sandstone were taken from the west wing of Panel 1 in Gaojiapu Coal Mine, Binchang Mining Area. Among them, coarse and medium sandstones were from the middle section of the Luhe Formation, while fine sandstone was from the lower section, with colors mainly light purplish red and light brownish red. ② Sample preparation: Using a Z3032X8 radial drilling machine and an SPQJ-300 slicing machine, the cores were processed into cylindrical samples with a diameter of 25 mm and a height of 50 mm. The coarse, medium, and fine sandstones were labeled as CSY, ZSY, and XSY, respectively, arranged from left to right as CSY, ZSY, and XSY. ③ Vacuum saturation: The samples were placed in a ZYB-II vacuum pressure saturation device, first vacuumed for 8 hours, then water was injected and pressurized to 8 MPa and maintained for more than 24 hours to ensure full saturation of pores. ④ NMR measurement: The samples were placed into a MacroMR12-150H-I low-field nuclear magnetic resonance instrument (Xijing University), and the equipment parameters were set as follows: 25 mm coil, CPMG sequence, O1 = 12773.43 Hz, P1 = 8 μs, P2 = 14.48 μs, SW = 250KHz, PRG = 2, TW = 1500 ms, NS = 8, TE = 0.06 ms, NECH echo number = 18000, to obtain the $T_2$ spectrum under saturated conditions. ⑤ Centrifugation treatment: The samples were placed in an H3-18K desktop high-speed centrifuge, set at 4000 r/min, and centrifuged for 15 minutes to remove movable water; then, they were tested again to obtain the $T_2$ spectrum of bound fluid.

### 2.2. CT Experiments

The detailed CT experimental steps are shown in Fig 2 and are as follows: ① Sample origin: identical to those used in the NMR test. ② Sample preparation: cores were first rough-cut with an SPQJ-300 diamond saw, further trimmed with a metallographic cutter, and finally machined into 5 mm × 5 mm × 15 mm rectangular prisms using an LC-200XP automatic high-precision cutter (Nabo, Jiaxing). From left to right, the nine specimens comprised three fine-, three medium-, and

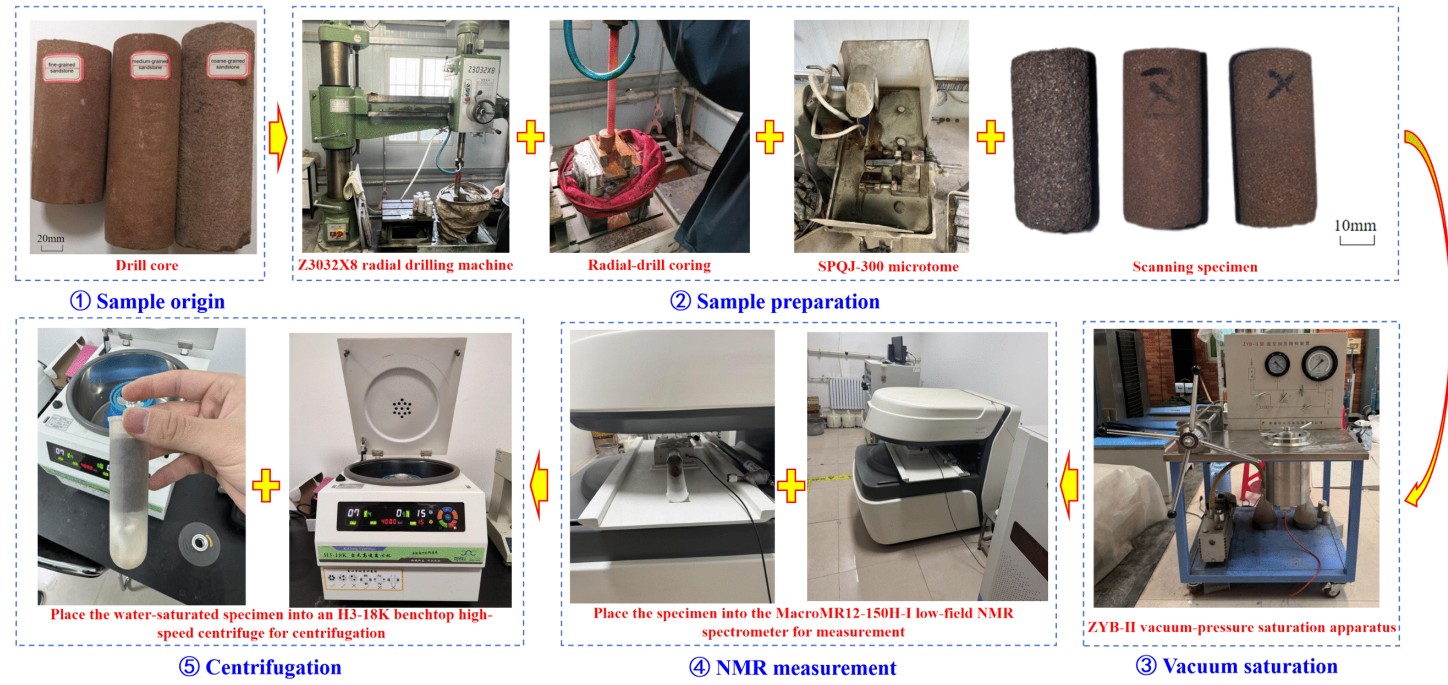

**Fig 1. NMR experimental procedure.**

three coarse-sandstone prisms; defect-free samples were selected for each lithotype and coded XSY, ZSY, and CSY, respectively. ③ Sample loadting: prisms were loaded into a Zeiss Xradia 510 Versa high-resolution 3D X-ray microscope (Guilin University of Technology) operated with (1) 3-D spatial resolution < 0.7 μm, (2) X-ray tube voltage 30–160 kV, maximum power 10 W, (3) 2k × 2k CCD camera, (4) interchangeable objective lenses (0.4X, 4X) manually selected according to resolution requirements, (5) detector travel range 290 mm, (6) propagation-based phase-contrast imaging, (7) maximum sample width 300 mm, and (8) maximum sample mass (including holder) 15 kg. ④ Test initiated: continuous transverse sections were scanned for each sandstone prism, yielding 1000 two-dimensional sequential images (TIFF format).

## 3. Analysis of sandstone 3D pore structure and seepage characteristics based on NMR technology

### 3.1. Analysis of $T_2$ spectrum characteristics

Fig 3 shows the $T_2$ spectra of the saturated and centrifuged samples of the three types of sandstone. As can be seen from Fig 3, the $T_2$ spectra of the saturated samples of coarse-grained, medium-grained, and fine-grained sandstone all exhibit multi-peak shapes, with 3, 5, and 4 peaks respectively. The total area of each $T_2$ spectrum and the starting point, peak point, endpoint, area, and peak proportion corresponding to each peak are shown in Table 1. In Table 1, CSY(B) represents the saturated sample of coarse-grained sandstone, and CSY(L) represents the centrifuged sample of coarse-grained sandstone, the same applies to the others, S represents the starting time of the peak, P represents the peak time, E represents the end time of the peak, M represents the peak area, R represents the peak proportion, and Z represents the total area. It can be seen from Table 1 that the total areas of the six $T_2$ spectra are 1196.31 ms², 1664.46 ms², 1672.91 ms², 390.35 ms², 1093.87 ms², and 1124.36 ms² respectively. For the CSY (B) sample, the S, P, E, M, and R of the first peak in the $T_2$ spectrum are 0.08 ms, 1.38 ms, 4.50 ms, 224.18 ms², and 18.74% respectively. The same applies to the other peaks, which will not be repeated here.

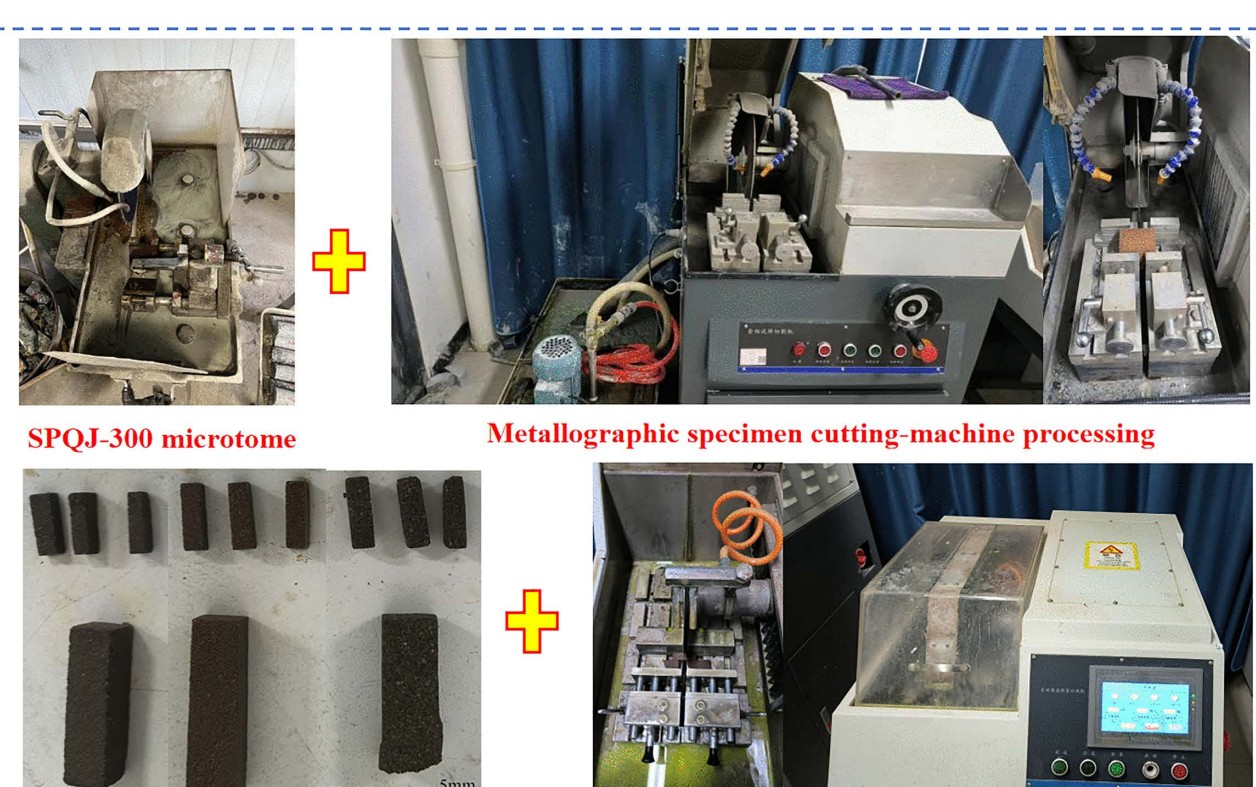

**① Sample preparation**

SPQJ-300 microtome

Metallographic specimen cutting-machine processing

Scanning specimen

LC-200XP automatic high-speed precision cutting machine

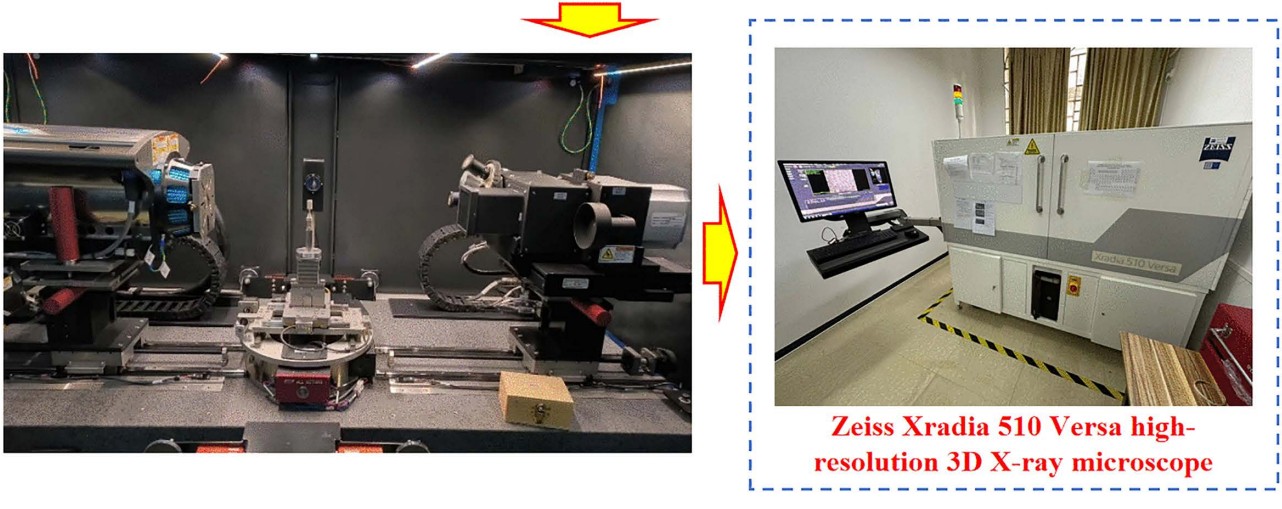

**② Sample loading**

**③ Test initiated**

Zeiss Xradia 510 Versa high-resolution 3D X-ray microscope

**Fig 2. CT experimental procedure.**

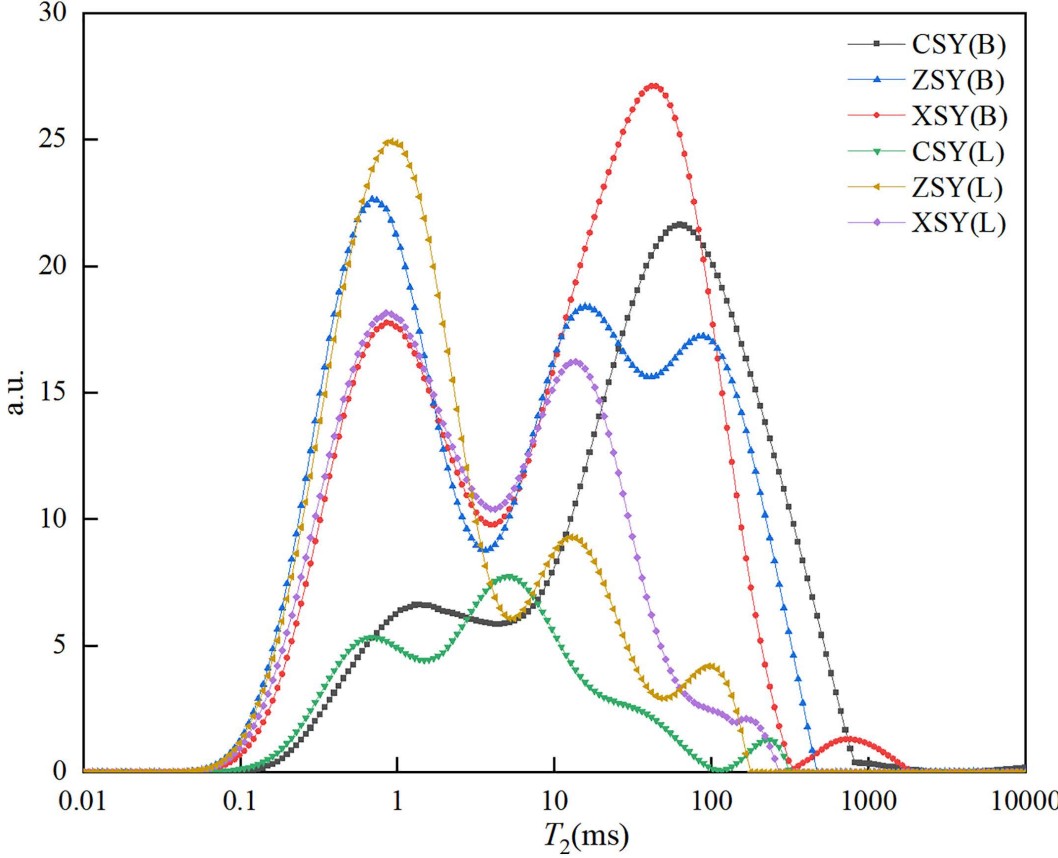

**Fig 3.** $T_2$ spectra of saturated and centrifuged samples of three types of sandstone.

As can be seen from Fig 3 and Table 1, the saturated samples have shorter peak starting times (0.02–0.008 ms), which are associated with smaller pores that allow water molecules to relax quickly. The longer peak ending times (3529.71–10000.00 ms) indicates the presence of larger pores, which allow water molecules to have a wider range of motion, thereby prolonging the relaxation time. Moreover, the saturated samples exhibit multiple peaks and a larger total area than the centrifuged samples, indicating a more complex pore structure in saturated sandstone, which includes movable fluid pores and bound fluid pores. In contrast, the $T_2$ spectra of the centrifuged samples are narrower, with fewer peaks (all three) and a smaller total area, indicating that the centrifugation process has removed the interconnected pores, leaving only the pores that restrict the movement of water molecules, namely the bound fluid pores.

### 3.2. Total porosity, bound fluid porosity, movable fluid porosity, dual T2c, and permeability

The values of total porosity, bound fluid porosity, movable fluid porosity, $T_{2c}$, and permeability are shown in Table 2. In Table 2, $\varphi_t$ represents the cumulative porosity, $\varphi_c$ represents the residual porosity, *BVI* represents the bound volume irreducible saturation (calculated by dividing $\varphi_c$ by $\varphi_t$), $\varphi_d$ represents the effective porosity (calculated by subtracting $\varphi_c$ from $\varphi_t$), *FFI* represents the free fluid saturation (calculated by dividing $\varphi_d$ by $\varphi_t$), $T_{2cutoff1}$ represents the first $T_2$ cutoff value, $T_{2cutoff2}$ represents the second $T_2$ cutoff value, and *k* represents the permeability. For CSY, ZSY, and XSY, the $\varphi_t$ values are 14.36%, 17.82%, and 16.09%, respectively. The $\varphi_c$ values are 4.51%, 11.62%, and 10.73%, respectively. The *BVI* values are 31.42%, 65.23%, and 66.69%, respectively. The $\varphi_d$ values are 9.85%, 6.20%, and 5.36%, respectively. The *FFI* values

**Table 1. Information of $T_2$ spectra of saturated and centrifuged samples of three types of sandstone.**

| Sample Number | Number of Peaks | S/ms | P/ms | E/ms | M/ms² | R/% | Z/ms² |
|---|---|---|---|---|---|---|---|
| CSY(B) | 1 | 0.08 | 1.38 | 4.50 | 224.18 | 18.74 | 1196.31 |
|  | 2 | 4.82 | 62.95 | 2673.84 | 971.08 | 81.17 |  |
|  | 3 | 4659.53 | 10000.00 | 10000.00 | 1.05 | 0.09 |  |
| ZSY(B) | 1 | 0.02 | 0.69 | 3.65 | 716.56 | 43.05 | 1664.46 |
|  | 2 | 3.92 | 15.70 | 41.50 | 527.59 | 31.70 |  |
|  | 3 | 44.49 | 89.07 | 471.38 | 419.66 | 25.21 |  |
|  | 4 | 1084.37 | 2673.84 | 3529.71 | 0.31 | 0.02 |  |
|  | 5 | 3783.46 | 4347.01 | 10000.00 | 0.33 | 0.02 |  |
| XSY(B) | 1 | 0.02 | 0.85 | 3.92 | 592.38 | 35.41 | 1672.91 |
|  | 2 | 4.20 | 41.50 | 333.13 | 1060.43 | 63.39 |  |
|  | 3 | 357.08 | 766.34 | 1889.65 | 20.08 | 1.20 |  |
|  | 4 | 7067.18 | 10000.00 | 10000.00 | 0.03 | 0.00 |  |
| CSY(L) | 1 | 0.02 | 0.69 | 1.48 | 125.07 | 32.04 | 390.35 |
|  | 2 | 1.59 | 5.17 | 117.59 | 254.76 | 65.26 |  |
|  | 3 | 126.04 | 235.43 | 333.13 | 10.52 | 2.70 |  |
| ZSY(L) | 1 | 0.02 | 0.91 | 5.17 | 820.17 | 74.98 | 1093.87 |
|  | 2 | 5.54 | 12.75 | 51.11 | 217.85 | 19.92 |  |
|  | 3 | 54.79 | 95.48 | 178.34 | 55.85 | 5.11 |  |
| XSY(L) | 1 | 0.02 | 0.85 | 4.20 | 636.72 | 56.63 | 1124.36 |
|  | 2 | 4.50 | 13.67 | 144.81 | 474.70 | 42.22 |  |
|  | 3 | 155.22 | 166.38 | 270.50 | 12.94 | 1.15 |  |

**Table 2. NMR parameters of three types of sandstone.**

| Core samples parameters | CSY | ZSY | XSY |
|---|---|---|---|
| $\varphi_t$/% | 14.36 | 17.82 | 16.09 |
| $\varphi_c$/% | 4.51 | 11.62 | 10.73 |
| BVI/% | 31.42 | 65.23 | 66.69 |
| $\varphi_d$/% | 9.85 | 6.20 | 5.36 |
| FFI/% | 68.58 | 34.77 | 33.31 |
| $T_{2cutoff1}$/ms | 1.70 | 9.66 | 19.34 |
| $T_{2cutoff2}$/ms | 16.83 | 20.73 | 31.44 |
| $k$/mD | 38.77 | 1.46 | 0.87 |

are 68.58%, 34.77%, and 33.31%, respectively. The $T_{2c}$ values are 16.83 ms, 20.73 ms, and 31.44 ms, respectively. The permeability ($k$) values are 38.77 mD, 1.46 mD, and 0.87 mD, respectively.

As can be seen from Table 2, CSY has the lowest cumulative porosity (14.36%) and the highest permeability (38.77 mD). It also has a relatively high movable fluid saturation (68.58%) and a relatively low bound fluid saturation (31.42%), which indicates that the CSY reservoir has a large capacity and good fluid mobility. In comparison, although ZSY has a slightly higher cumulative porosity (17.82%) than CSY, its permeability (1.46 mD) and effective porosity (6.20%) are lower, and its bound fluid saturation (65.23%) is higher. This means that its reservoir capacity is better than that of CSY, but its fluid mobility is worse than that of CSY. The cumulative porosity of XSY (16.09%) is similar to that of CSY, but its

permeability (0.87 mD) and effective porosity (5.36%) are the lowest, and its bound fluid saturation (66.69%) is the highest, indicating that its reservoir capacity is the best, but its fluid mobility is the worst.

The specific methods for obtaining the corresponding values are as follows: ①Total porosity and bound fluid porosity: The low-field nuclear magnetic resonance $T_2$ spectra of different porosity standard samples (Fig 4) were measured using the low-field nuclear magnetic resonance measuring instrument. The relationship between the nuclear magnetic resonance-calculated porosity and the integral area of the $T_2$ spectrum was established based on the known porosity of the standard samples. Then, the corresponding calculations of porosity were performed on the $T_2$ spectra of the tested rock samples. The calibrated calculation formula is shown in Eq (1).

$$\varphi_{NMR} = \left(\frac{S}{V} - 3.65\right)/14.21$$

(1)

In the formula, $S$ represents the integral area of the $T_2$ spectrum (dimensionless), which reflects the total fluid relaxation characteristics of the sample; $V$ refers to the volume of the sample (unit: cm³), which is used to normalize the porosity parameter; $\varphi_{NMR}$ is the porosity percentage (%) calculated by low – field nuclear magnetic resonance (LF – NMR) technology;

②Movable fluid porosity: The $T_2$ spectra of the three types of sandstone under saturated and bound water conditions were accumulated to obtain the cumulative porosity curves. The maximum values of the cumulative porosity represent the total porosity and residual porosity, respectively. The effective porosity of the three types of sandstone can be obtained by subtracting the residual porosity from the total porosity.

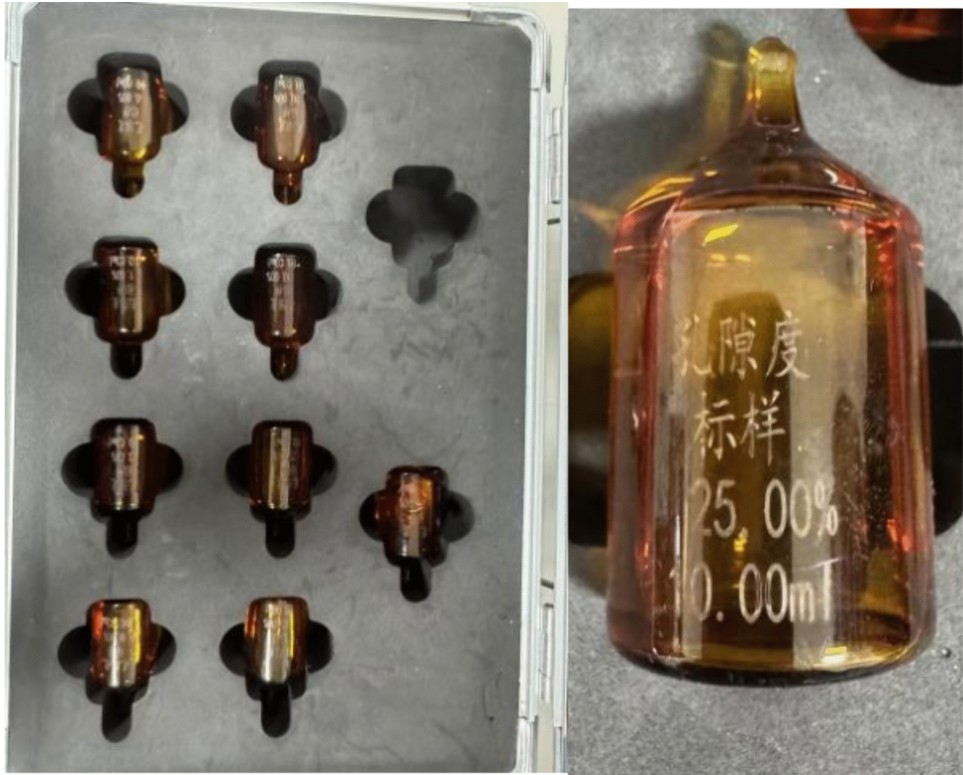

**Fig 4. Standard samples with different porosities.**

③Dual $T_2$ cutoff values: The classic NMR cutoff model uses a single $T_2$ cutoff value ($T_{2cutoff2}$) to divide the core porosity into two parts, assuming that the fluid in pores larger than the cutoff value is fully movable and that in pores smaller than the cutoff value is fully bound [32]. However, experimental results show that after centrifugation, the amplitude of the $T_2$ spectrum of pores smaller than $T_{2c}$ is less than that before centrifugation, indicating that the fluid in pores smaller than $T_{2c}$ still has some mobility. Therefore, a dual-cutoff approach was used to classify pores into three types: fully movable (larger than $T_{2cutoff2}$), fully bound (smaller than $T_{2cutoff1}$), and partially movable (larger than $T_{2cutoff1}$ but smaller than $T_{2cutoff2}$). The second cutoff value $T_{2cutoff2}$ was determined by drawing a horizontal line from the bound fluid porosity to the left, and the intersection with the cumulative porosity curve of the saturated sample gives the $T_{2cutoff2}$ value. The selection principle of the first cutoff value $T_{2cutoff1}$ is shown in Eq (2).

$$\frac{\varphi_{NMR,T2cutoff} - \varphi_{wi,T2cutoff}}{\varphi_{NMR} - \varphi_{wi}} > 0.01 \tag{2}$$

In the formula, $\varphi_{NMR}$ and $\varphi_{wi}$ are the NMR porosities before and after centrifugation, respectively. $\varphi_{NMR,T2cutoff}$ and $\varphi_{wi,T2cutoff}$ are the cumulative NMR porosity curve values at the abscissa $T_{2cutoff}$ before and after centrifugation. The first cutoff value $T_{2cutoff1}$ is the smallest $T_{2cutoff}$ that satisfies Eq (2). The dual $T_{2c}$ and effective porosity of the three types of sandstone are shown in Fig 5–7.

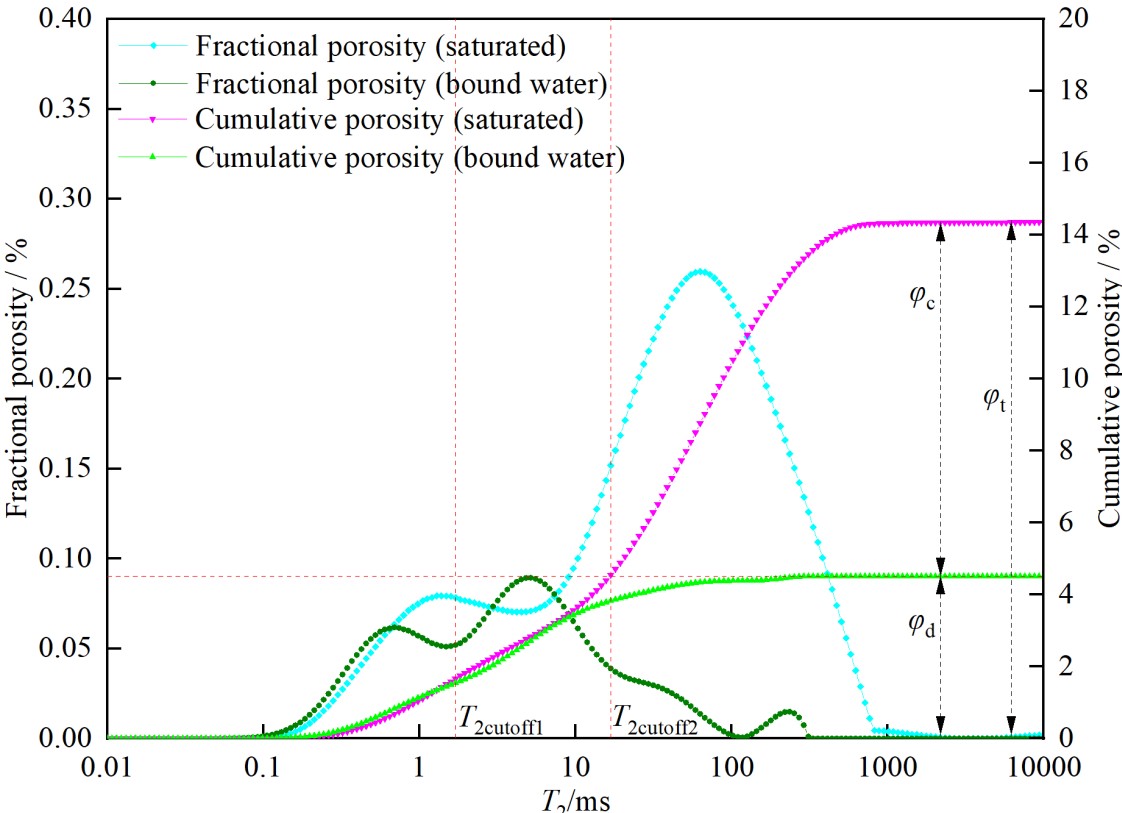

**Fig 5. Dual $T_{2c}$ and effective porosity of coarse sandstone.**

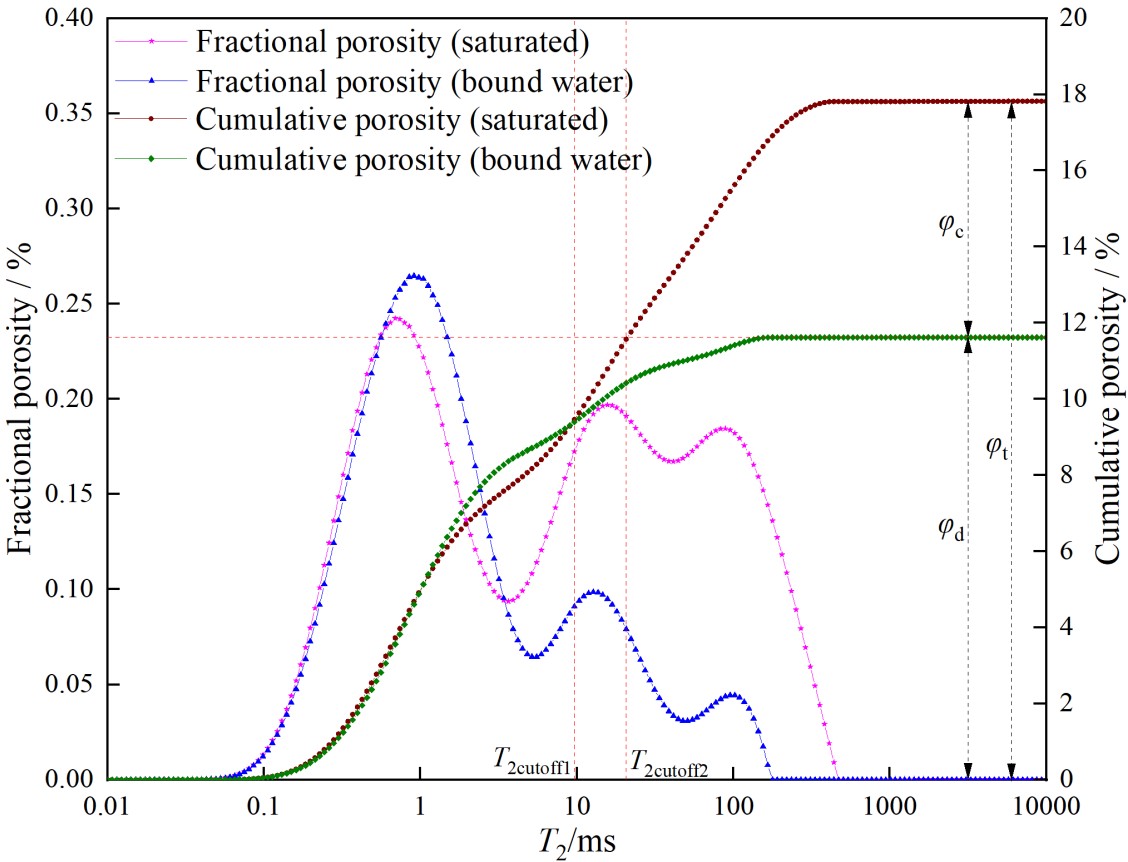

**Fig 6. Dual $T_{2c}$ and effective porosity of medium sandstone.**

④Permeability: Permeability calculation is another important application of nuclear magnetic resonance (NMR) technology. Several mainstream NMR permeability models have been widely used, including the classic Coates model (Eq (3)), its extended version (Eq (4)), the SDR model (Eq (5)), and the further developed SDR – REV extended model (Eq (6)). These models, with their unique algorithms and high accuracy, play an important role in reservoir evaluation. The formulas are as follows:

$$K_{\text{Coates–cutoff}} = \left( \frac{\varphi_{\text{NMR}}}{C_1} \right)^4 \left( \frac{FFI}{BVI} \right)^2 \tag{3}$$

$$K_{\text{Coates–cutoff–REV}} = \left( \frac{\varphi_{\text{NMR}}}{C_2} \right)^m \left( \frac{FFI}{BVI} \right)^n \tag{4}$$

$$K_{\text{SDR}} = C_3 \left( \frac{\varphi_{\text{NMR}}}{100} \right)^2 T_{2g}^2 \tag{5}$$

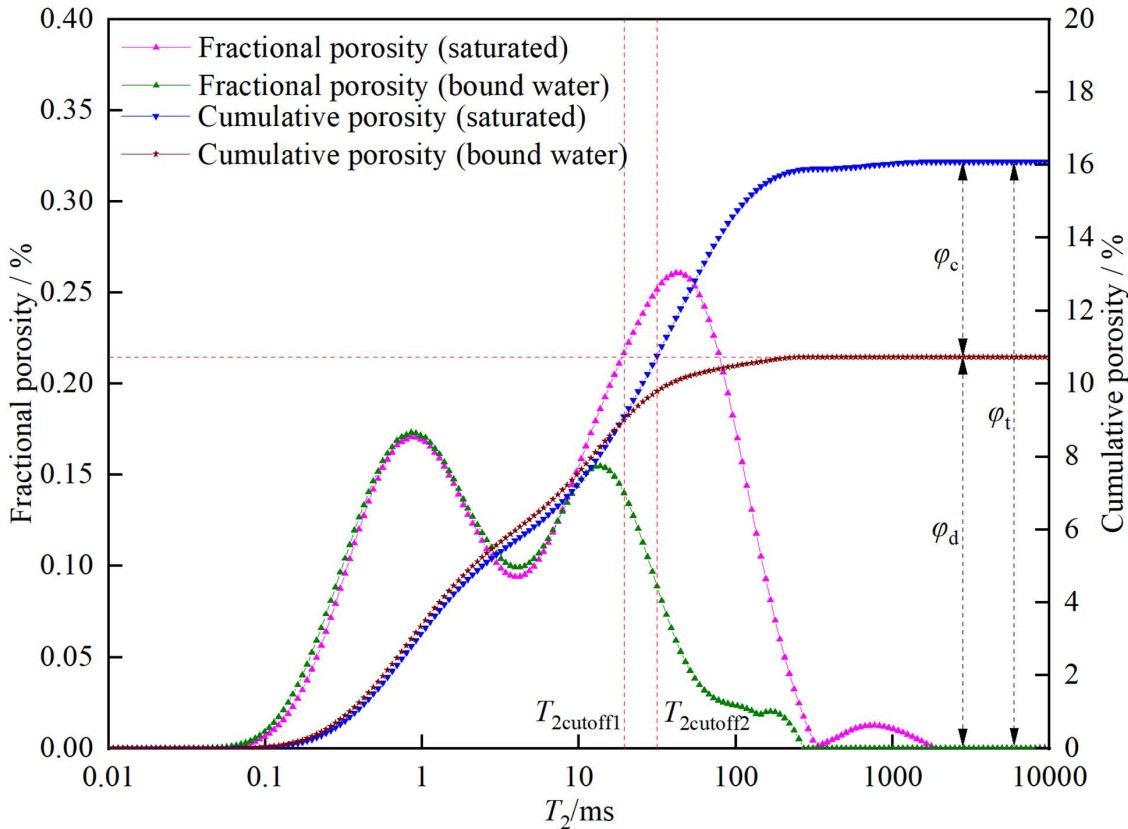

**Fig 7. Dual $T_{2c}$ and effective porosity of fine sandstone.**

$$K_{SDR-REV} = C_4 \left( \frac{\varphi_{NMR}}{100} \right)^m T_{2g}^n \tag{6}$$

In the formulas, $K_{coates-cutoff}$ (mD), $K_{coates-cutoff-REV}$ (mD), $K_{SDR}$ (mD), and $K_{SDR-REV}$ (mD) represent the Coates permeability, the extended Coates model permeability, the SDR permeability, and the SDR extended permeability, respectively. $\phi_{NMR}$ (%) is the NMR calculated porosity after sample saturation with water. $FFI$ (%) is the movable water saturation. $BVI$ (%) is the bound water saturation. $T_{2g}$ (ms) is the geometric mean of the $T_2$ spectrum. $C_n$, m, and n are model constants.

The geometric mean of the NMR relaxation time $T_{2g}$ is shown in Eq (7).

$$T_{2g} = {}^{\varphi_{NMR}}\!\sqrt{T_{21}^{\varphi_1} T_{22}^{\varphi_2} T_{23}^{\varphi_3} \dots T_{2i}^{\varphi_i}} \tag{7}$$

In the formula, $T_{2g}$ is the geometric mean, $\varphi_{NMR}$ is the NMR porosity of the rock, $T_{2i}$ are the individual $T_2$ distribution points, and $\varphi_i$ are the porosity components corresponding to each $T_2$ distribution point.

The specific parameters of the permeability models are shown in Table 3 [33]. Among them, the correlation coefficient R between Coates – cutoff – REV and air – permeability is the highest. Therefore, it is used to calculate the permeability of the three types of sandstone in this study.

**Table 3. Parameters of permeability models.**

| Permeability Model | Permeability Formulas | Coefficient | Correlation coefficient R with air-permeability |
|---|---|---|---|
| Coates-cutoff | $K_{Coates-cutoff} = \left(\frac{\varphi_{NMR}}{C_1}\right)^4 \left(\frac{FFI}{BVI}\right)^2$ | $C_1 = 9.10$ | 0.87 |
| Coates-cutoff-REV | $K_{Coates-cutoff-REV} = \left(\frac{\varphi_{NMR}}{C_2}\right)^m \left(\frac{FFI}{BVI}\right)^n$ | $C_2 = 9.12$ $m = 3.21$ $n = 2.82$ | 0.92 |
| SDR | $K_{SDR} = C_3 \left(\frac{\varphi_{NMR}}{100}\right)^2 T_{2g}^2$ | $C_3 = 111.54$ | 0.72 |
| SDR-REV | $K_{SDR-REV} = C_4 \left(\frac{\varphi_{NMR}}{100}\right)^m T_{2g}^n$ | $C_4 = 85.76$ $m = 2.26$ $n = 0.85$ | 0.82 |

### 3.3. Pore distribution

According to nuclear magnetic resonance (NMR) theory, the NMR $T_2$ spectrum of rock saturated with a single-phase fluid can reflect the size and proportion of its internal pores [34]. Based on the principle of measuring capillary pressure curves by mercury intrusion porosimetry (MIP), the size of the pore-throat and the distribution of the connected pore volume can be obtained from the capillary pressure curve. The NMR $T_2$ spectrum of the water-saturated core can also evaluate the size of the pores and the corresponding pore volume distribution. Since the pore distribution reflected by the two measurement methods is the same, establishing the relationship between $T_2$ relaxation time and throat radius can convert the NMR $T_2$ distribution into the pore-throat radius distribution. For water-saturated sandstone, the relationship between relaxation time and throat radius r [35] is shown in Eq (8).

$$T_2 = \frac{(c_1 r)^n}{\rho_2 F_s}$$

(8)

In Eq (8), $c_1$ is the average pore-to-throat ratio; $n$ is the power-law exponent; $F$s is the pore shape factor; $\rho_2$ is the transverse surface relaxation intensity, which depends on the size and saturation fluid properties of the pore surface properties and mineral composition, in units of µm/ms.

Letting $C = \frac{(\rho_2 F_s)^{1/n}}{c_1}$, Eq (8) can be further transformed into

$$r = C T_2^{1/n}$$

(9)

Therefore, based on the combined testing of mercury intrusion and nuclear magnetic resonance (NMR), the values of $C$ and $n$ can be obtained, which allows the conversion of the $T_2$ relaxation time distribution of water-saturated cores into pore-throat radius distribution curves. Referring to the research results of Feng Longfei et al. [6] on converting the NMR $T_2$ spectrum of Luhe Formation sandstone from the first mining district of Gaojiabao Coal Mine into pore distribution: $r = 0.01933 T_2^{0.6418}$, the $T_2$ spectrum distribution curve is converted into a pore distribution curve. Furthermore, referring to the pore classification method in reference [36], pores with $T_2$ relaxation times between 0.01–1 ms are defined as micropores, pores with $T_2$ relaxation times between 1–10 ms are defined as small pores, pores with $T_2$ relaxation times between 10–100 ms are defined as medium pores and pores with $T_2$ relaxation times between 100–10000 ms are defined as large pores and micro-fractures (Fig 8).

The pore-fracture structure of the three types of sandstone is the storage space and transport channel for water in sandstone. Fig 8 shows the pore diameter distribution of the three types of sandstone. To further investigate the characteristics of the pore-fracture structure of the three types of sandstone, a quantitative analysis was made of the proportions of pores of

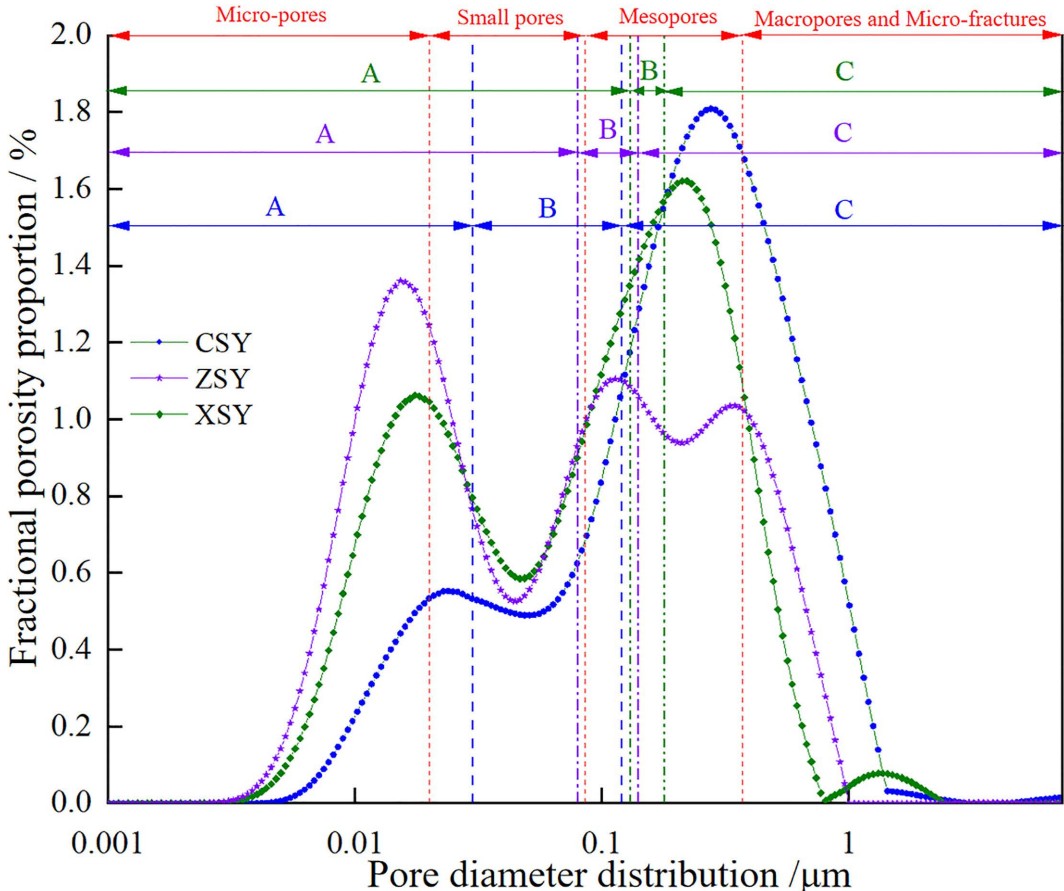

**Fig 8. Pore diameter distribution.**

different sizes (micropores, small pores, medium pores, and large pores (and micro – fractures)) and different types of pores (fully bound, partially movable, and fully movable pores, represented by A, B, and C respectively, where A is the proportion of pores with $T_2$ values less than $T_{2cutoff1}$, B is the proportion of pores with $T_2$ values between $T_{2cutoff1}$ and $T_{2cutoff2}$, and C is the proportion of pores with $T_2$ values greater than $T_{2cutoff2}$) in the three types of sandstone. The results are shown in Table 4. It can be seen from the table that the proportions of micropores, small pores, medium pores, and large pores (and micro-fractures) in CSY are 7%, 18%, 48%, and 27% respectively, those in ZSY are 28%, 26%, 34%, and 12% respectively, and those in XSY are 19%, 27%, 46%, and 8% respectively. The proportions of A, B, and C in CSY are 12%, 20%, and 68% respectively, those in ZSY are 45%, 13%, and 42% respectively, and those in XSY are 64%, 7%, and 29% respectively. The proportions of pores of different sizes in the three types of sandstone are shown in Fig 9. It can be seen from Fig 9 that the proportion

**Table 4. Proportion of various pores in three types of sandstone.**

| Sandstone specimen | Micropores /% | Small Pores /% | Medium pores /% | Large pores and micro-fractures /% | A /% | B /% | C /% |
|---|---|---|---|---|---|---|---|
| CSY | 7 | 18 | 48 | 27 | 12 | 20 | 68 |
| ZSY | 28 | 26 | 34 | 12 | 45 | 13 | 42 |
| XSY | 19 | 27 | 46 | 8 | 64 | 7 | 29 |

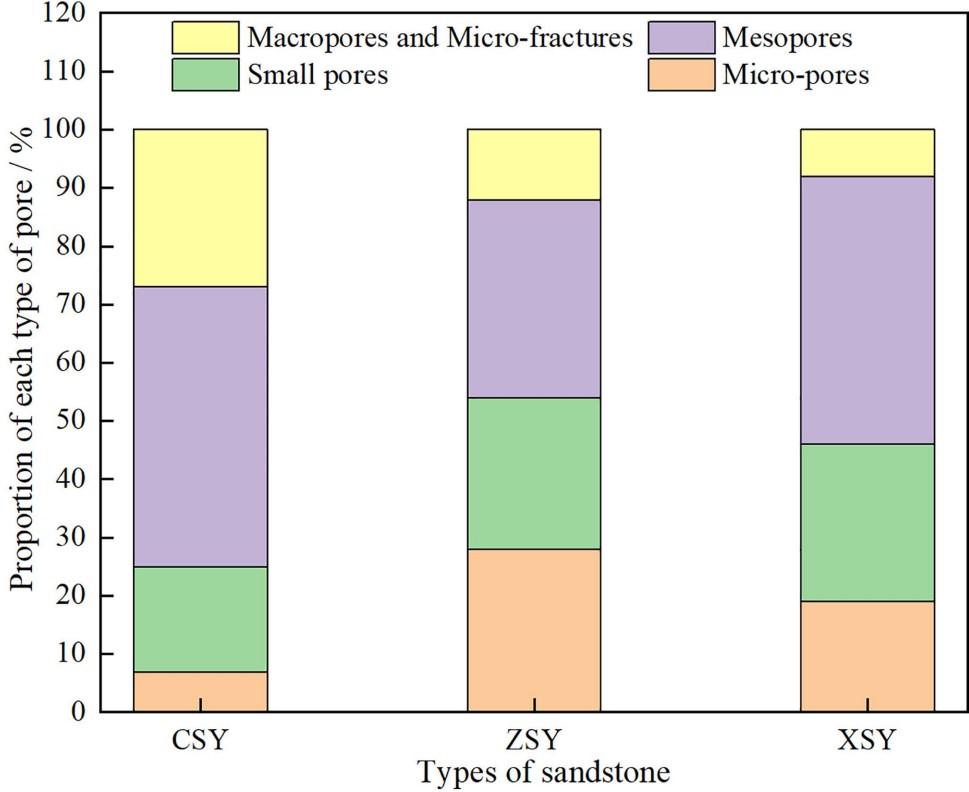

**Fig 9. Proportion of various pores classified by size in three types of sandstone.**

of medium pores is generally high in the three types of sandstone, while the proportions of large pores micro-fractures, and small pores are relatively low. The proportion of large pores and micro-fractures is the highest in CSY, reaching 27%, while it is only 12% and 8% in ZSY and XSY respectively. This indicates that the permeability of CSY is the highest, as large pores and micro-fractures are usually associated with higher permeability. Furthermore, the proportions of pores of different types in the three types of sandstone are shown in Fig 10. It can be seen from Fig 10 that the proportion of fully movable pores is the highest in CSY, reaching 68%, while it is only 42% and 29% in ZSY and XSY respectively. This further confirms that the permeability of coarse-grained sandstone is the best. It can also be seen that the proportion of fully bound pores is the highest in XSY, reaching 64%, indicating that XSY has the best storage capacity.

### 3.4. Fractal characteristic analysis of rock samples

The fractal theory provides an effective tool for revealing the complex and multi-scale pore-fracture structure of sandstone. It is of great significance for quantifying the heterogeneity, size distribution, and structural complexity of pore structures. The fractal dimension can quantitatively characterize the complexity of pore-fracture structures in porous media. The larger the fractal dimension, the more complex the pore-fracture structure of the medium. The formula for calculating the fractal dimension using the low-field nuclear magnetic resonance (NMR) $T_2$ spectrum is given in reference [37] as:

$$S_v = \left( \frac{T_{2max}}{T_2} \right)^{D-3}$$

(10)

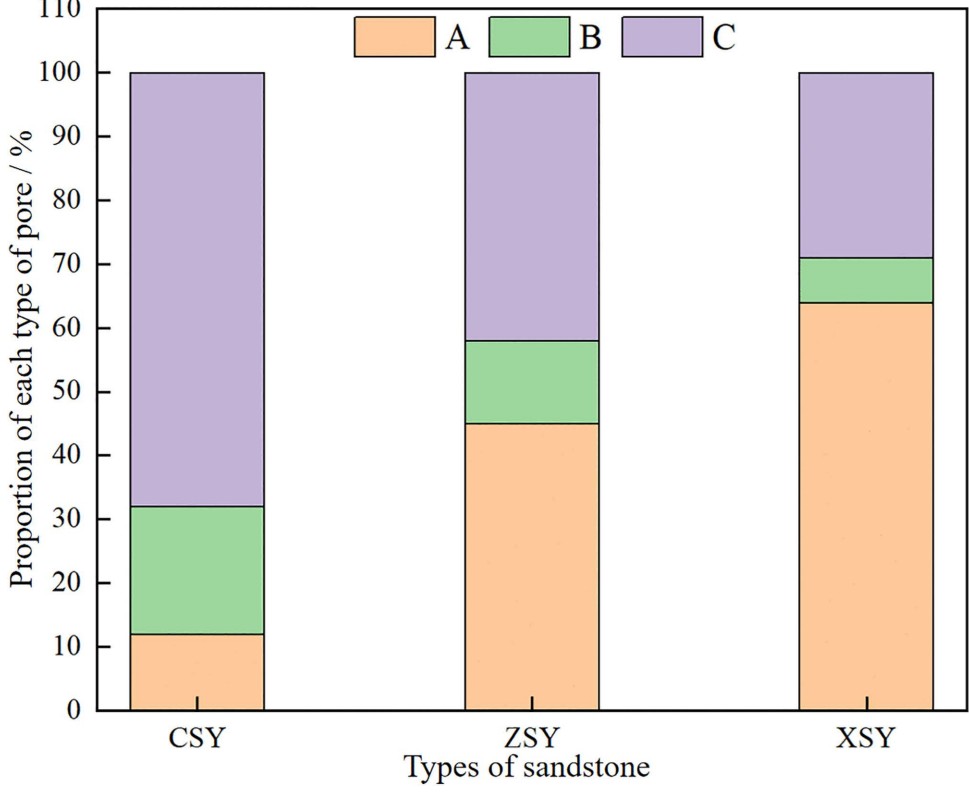

**Fig 10. Proportion of various pores classified by type in three types of sandstone.**

In Eq (10): $S_v$ is the percentage of the cumulative pore volume with transverse relaxation time less than $T_2$ of the total pore volume, in percent (%); $D$ is the fractal dimension; $T_{2max}$ is the maximum transverse relaxation time.

By integrating both sides of Eq (10) and rearranging the terms, we can obtain:

$$\lg S_v = (3 - D)\lg T_2 + (D - 3)\lg T_{2max} \tag{11}$$

Therefore, if the pore structure of the rock sample has fractal characteristics, the fractal dimension can be calculated through the relationship between $\lg S_v$ and $\lg T_2$, using the slope $\gamma$ of the plot of $\lg S_v$ versus $\lg T_2$. The fractal dimension is given by:

$$D = 3 - \gamma \tag{12}$$

The core pores were divided into two parts using $T_{2cutoff2}$ as the boundary. It was assumed that the fractal dimension of pores larger than the cutoff value was the fractal dimension of connected pores, while the fractal dimension of pores smaller than the cutoff value was the fractal dimension of isolated pores. The fractal dimensions of the three types of sandstone were calculated respectively, as shown in Fig 11–13.

The curves in Fig 11–13 were linearly fitted using Origin to obtain their slopes, and the fitting results were shown in each Figure. The fractal dimensions of the connected and isolated pores in the three types of sandstone were calculated using Eq (12) and are presented in Table 5.

As can be seen from Fig 11–13, the curves exhibit a distinct two-segment characteristic and can be well described by a linear relationship, indicating that the pore structures of the three types of sandstone have fractal characteristics. It can be seen from Table 5 that, by comparing the fractal dimensions of the pores of the three types of sandstone, the fractal dimension of the connected pores of XSY is the largest, indicating that the distribution of large pores and micro-fractures in XSY is complex and highly heterogeneous, which is not conducive to fluid seepage, while CSY is the most conducive to fluid flow. Since CSY has a low content of micropores, there are more values with a signal strength close to 0, leading to infinite values when taking the logarithm. Therefore, many values with a strength of 0 were discarded, resulting in a larger fractal dimension for the isolated pores of CSY. Thus, we only compare the fractal dimensions of the isolated pores of ZSY and XSY and find that the fractal dimension of the isolated pores of XSY is larger, indicating that the distribution of micro-pores and small pores in XSY is more complex and highly heterogeneous, which is more conducive to fluid storage. This is consistent with the previous analysis. Therefore, the pore structures of the three types of sandstone and their control on fluid storage and seepage have been analyzed from multiple perspectives.

## 4. Analysis of sandstone 3D pore structure and seepage simulation based on X-CT technology

### 4.1. Identification of 3D pore structure characteristics and seepage simulation analysis process

In this study, the three-dimensional reconstruction of the X-CT digital images of the three types of sandstone was comprehensively carried out using Avizo, Netfabb, and Comsol technologies. Pore network models were established, and three-dimensional pore structure characteristic identification and seepage simulation research were conducted.

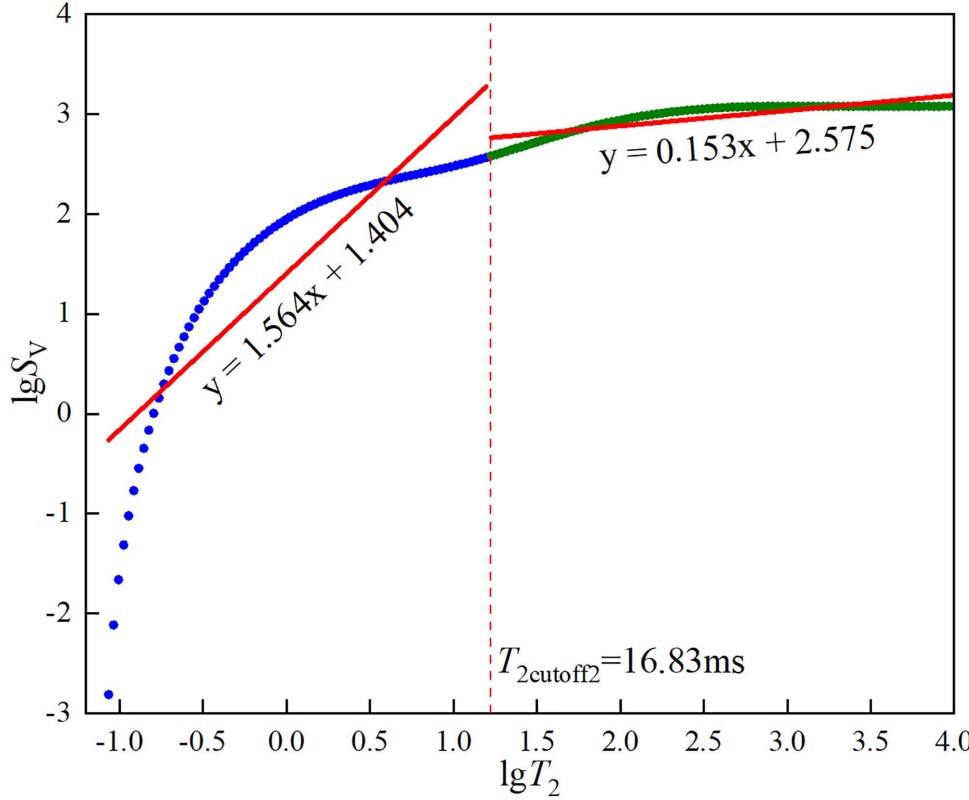

**Fig 11. Fitting results of lg$S_v$ against lg$T_2$ for coarse sandstone.**

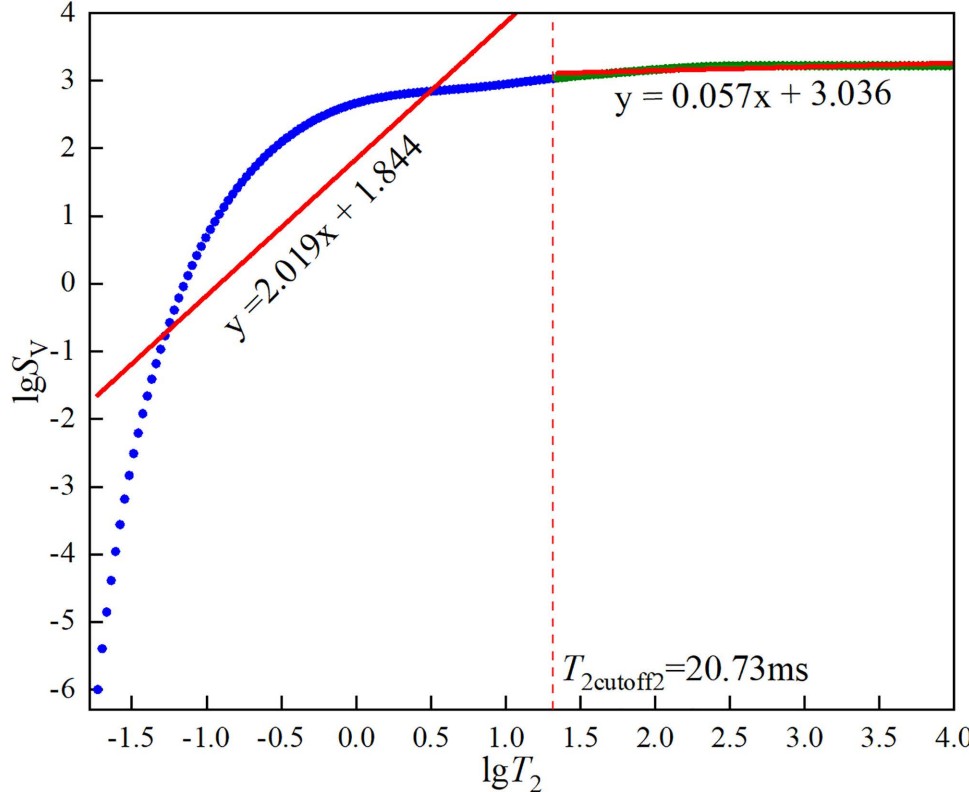

**Fig 12. Fitting results of lg$S_v$ against lg$T_2$ for medium sandstone.**

The specific analysis process is shown in Fig 14: ①Resolution setting, target area selection, and noise reduction. The 1000 CT two - -two-dimensional slices with a resolution of 5μm obtained from the X-CT scanning of each sandstone were imported into Avizo. The resolution was set to 5μm first. Then, the target area was selected, and the median filter was used to filter and reduce noise in the target area. ②Determination of the threshold segmentation method and pore extraction. Pores were extracted using a combined segmentation method of Interactive Thresholding (for selecting larger pores) and Top – hat (for selecting micro-fractures) [38], and the porosity was calculated. ③Calculation of total porosity, isolated and connected porosity, and fractal dimension. Fractal dimension can be used to quantitatively describe the complexity of pore structure [39,40]. ④Establishment of a pore network model based on connected pores and calculation of pore radius, pore-throat radius, permeability, coordination number, and tortuosity. ⑤Seepage numerical simulation. The connected pore models of the coarse-grained, medium-grained, and fine-grained sandstone samples were subjected to one-directional peristaltic flow seepage simulation in the Z-axis direction in Comsol. First, the connected pore models were meshed and optimized using Avizo and Netfabb, and the optimized surface meshes were saved as STL-format files. Then, the STL files were imported into the peristaltic flow module of Comsol Multiphysics for volume meshing. The quality of the volume mesh was checked. When the mesh quality met the requirements, the material properties and boundary conditions were set. To ensure the accuracy of the simulation as much as possible, the inlets and outlets of the imported models were carefully selected to ensure that all of them were selected. In this study, the inlets and outlets were carefully selected using a frame-selection method, and then a seepage simulation analysis was carried out.

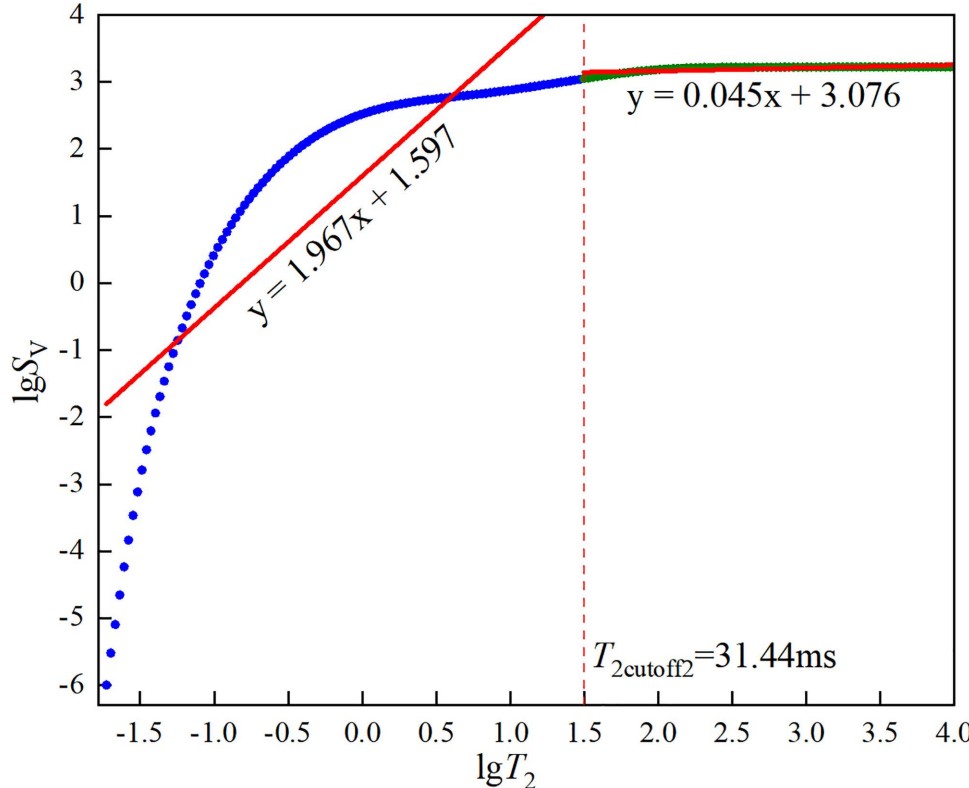

**Fig 13. Fitting results of lg$S_v$ against lg$T_2$ for fine sandstone.**

**Table 5. Fractal dimensions of connected and isolated pores in three types of sandstone.**

| Sandstone specimen | $D_1$ | $D_2$ |
|---|---|---|
| CSY | 1.436 | 2.847 |
| ZSY | 0.981 | 2.943 |
| XSY | 1.033 | 2.955 |

## 4.2. Three-dimensional pore structure characteristics of different sandstones

**4.2.1. Visualization of 3D pore structure.** Based on the 3D pore structure characteristic identification method and process (Fig 14), a visualization analysis of the pore space structure characteristics of the three types of sandstone was carried out. The total porosity of the sandstone samples was divided into connected and unconnected porosity. Connected porosity is the core of the sandstone's storage and permeability performance, while unconnected porosity only contributes to the storage capacity. It can be easily seen from the analysis results in Fig 15 (CSY, ZSY, and XST on the left of the Figure represent coarse-grained, medium-grained, and fine-grained sandstone respectively. The Figures from left to right are the total pore structure, isolated pore structure, connected pore structure, segmented connected pores, and pore network model respectively) the transparent part is the matrix, the blue part is the connected pores, and the purple part is the unconnected pores. The unconnected pores are mainly isolated pores and a small amount of locally connected pores. At the same time, based on the segmentation of the connected pores of the three types of sandstone, an equivalent pore

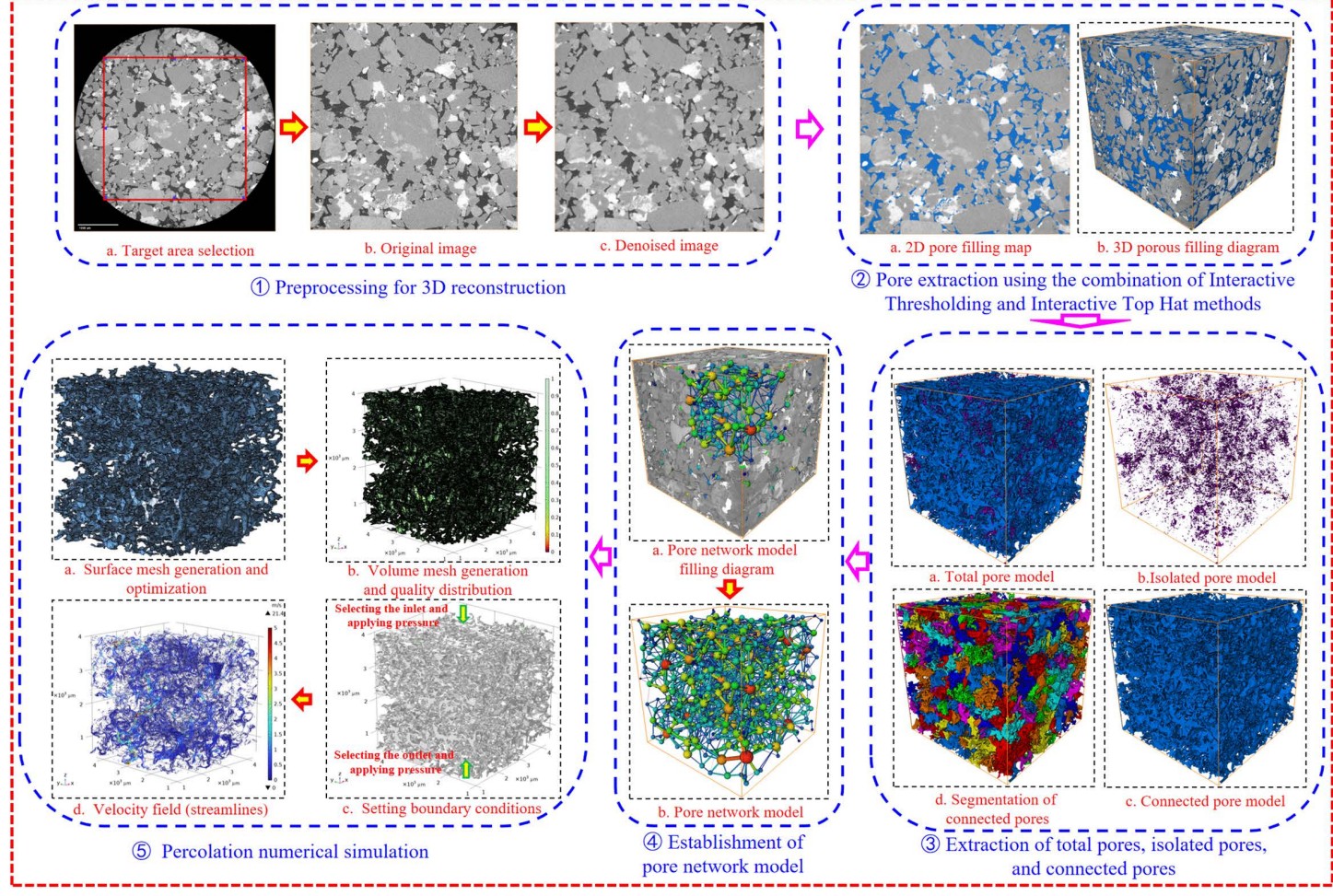

**Fig 14. Workflow for identification of 3D pore structure characteristics and seepage simulation analysis.**

network model with pore morphology topological structure, that is the "sphere-tube" model, was constructed. It can be seen from the results of the pore network model extraction that the pore network (spheres represent pores and tubes represent throats) fits well with the pore-throat structure of the CT-scanned core, and the extraction results of the pore network model are reliable.

**4.2.2. Quantitative characterization of 3D pore structure.** Total porosity ($\varphi_f$), connected porosity ($\varphi_c$), connectivity ($L$) (calculated according to the formula [41] $L = \varphi_c/\varphi_f$), and total pore fractal dimension ($D_f$) were extracted from the three-dimensional reconstructed models of different sandstone samples as basic pore structure parameters. Meanwhile, pore diameter distribution ($D_p$) and its mean value ($\overline{D}_p$), pore-throat radius distribution ($r_t$) and its mean value ($\overline{r}_t$), and absolute permeability ($k$) ($k$ were calculated based on single-directional laminar flow. Based on the pore network model, assuming the fluid is incompressible, the volume flow rate was first calculated according to Poiseuille's principle, and then $k$ was calculated according to Darcy's law [42]), tortuosity ($T$) (Tortuosity of sandstone refers to the ratio of the actual path length traveled by fluid particles in the sandstone pore structure to the geometric length of the sandstone medium. The value of tortuosity is usually greater than 1. When the pore channel is straight, tortuosity equals 1. The more tortuous the pore channel is, the greater the tortuosity.), coordination number distribution ($\zeta$) and its mean value ($\overline{\zeta}$) (In the sandstone pore

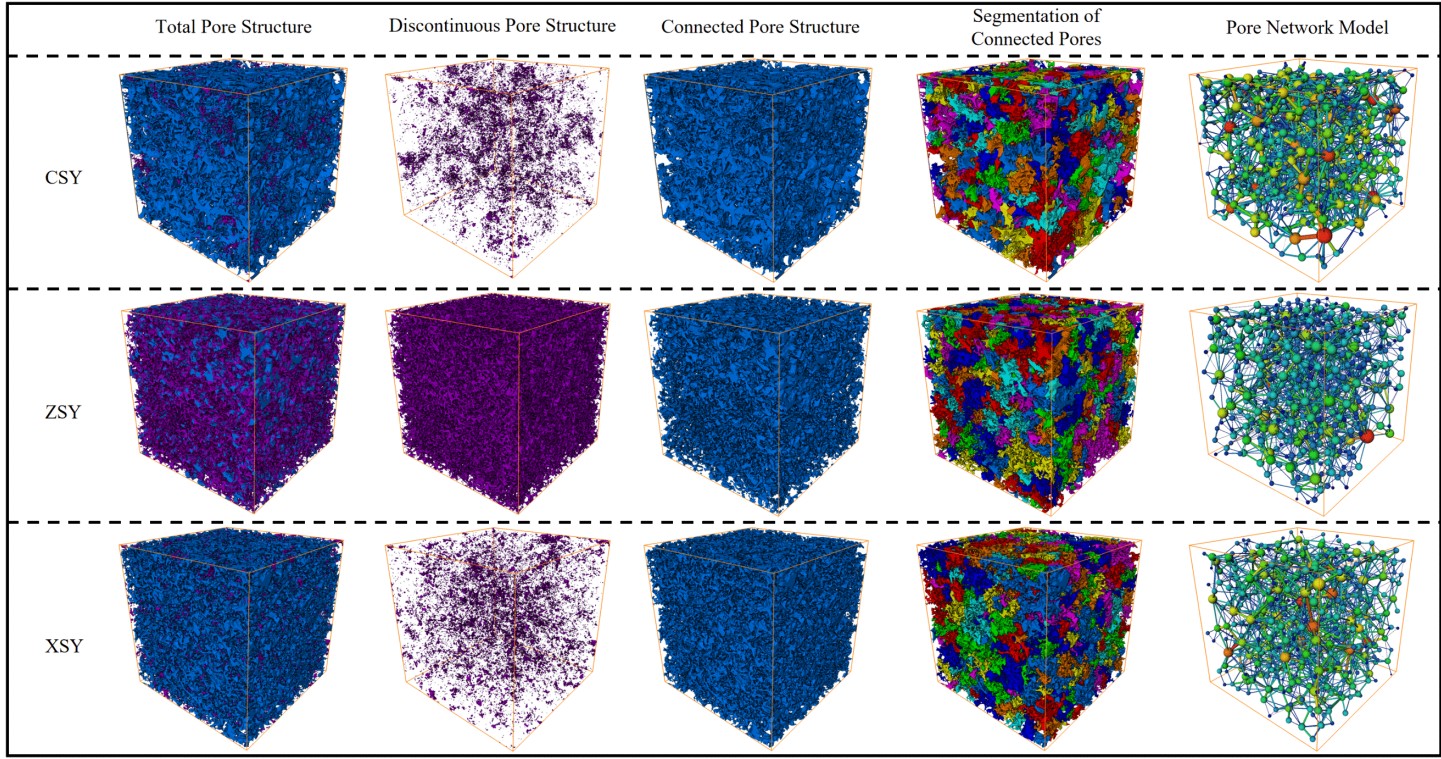

**Fig 15. Three-dimensional reconstruction and pore network models of three types of sandstone samples.**

network model, the average coordination number ($\zeta$) is a key parameter, which describes the average number of throats connected to each pore in the pore network.) were extracted from the equivalent pore network model as pore network topological parameters for quantitative analysis of pore structure characteristics. The specific data are shown in Table 6.

As can be seen from the data in Table 6, the CSY sample has the highest total porosity (16.43%), connected porosity (16.19%), and connectivity (98.54%), indicating the highest degree of pore development and good connectivity. Its total pore fractal dimension (1.53) is also the largest, meaning the highest complexity of pore space. The ZSY sample has the second-highest total porosity (15.51%), connected porosity (15.21%), and connectivity (98.07%), with a total pore fractal dimension (1.51) in the middle. The XSY sample has the lowest total porosity (14.68%), connected porosity (14.36%), and connectivity (97.82%), as well as the smallest total pore fractal dimension (1.50), indicating the lowest degree of pore development and space complexity. In terms of pore diameter distribution, the CSY sample has the largest mean pore diameter (17.50μm) and mean pore-throat radius (32.63μm), indicating larger pore and pore-throat sizes that are conducive to fluid flow. The XSY sample has the smallest mean pore diameter (17.03μm) and mean pore-throat

**Table 6. Pore structure and pore network topological features of different sandstone samples.**

| | $\varphi_f$ / % | $\varphi_c$ / % | $L$ / % | $D_f$ | $D_p$ /μm | $\overline{D}_p$ /μm | $r_t$ /μm | $\overline{r}_t$ /μm | $k$/D | $T$ | $\zeta$ | $\overline{\zeta}$ |
|---|---|---|---|---|---|---|---|---|---|---|---|---|
| CSY | 16.43 | 16.19 | 98.54 | 1.53 | 6.20~508.64 | 17.50 | 1.33~119.76 | 32.63 | 10.88 | 1.63 | 1~28 | 8.05 |
| ZSY | 15.51 | 15.21 | 98.07 | 1.51 | 6.20~495.87 | 17.16 | 1.33~111.80 | 29.11 | 10.56 | 1.65 | 1~25 | 7.83 |
| XSY | 14.68 | 14.36 | 97.82 | 1.50 | 6.20~486.50 | 17.03 | 1.33~100.18 | 28.75 | 8.40 | 1.69 | 1~23 | 7.65 |

radius (28.75µm), resulting in relatively higher flow resistance. Regarding permeability, the CSY sample has the highest absolute permeability (10.88 D), indicating the strongest fluid flow capacity. The XSY sample has the lowest absolute permeability (8.40 D), with the weakest flow capacity. In terms of tortuosity, the CSY sample has the smallest tortuosity (1.63), meaning the simplest fluid flow path. The XSY sample has the largest tortuosity (1.69), with the most complex flow path. In terms of coordination number, the CSY sample has the largest mean value (8.05), indicating the most complex pore network connections. The XSY sample has the smallest mean value (7.65), with the simplest network connections.

## 5. Seepage mechanism and numerical simulation

Pore-scale flow physics governs fluid transport in sandstone reservoirs. Here we focus on how the reconstructed pore architecture controls single-phase flow and simulate it with a creeping-flow model (Navier–Stokes equations).

### 5.1. Fundamental control of pore structure on seepage mechanisms

The pore structure of sandstone (pore-size distribution, connectivity, topological morphology, and fractal characteristics) directly dominates the differences in seepage mechanisms by altering flow paths, resistance, and fluid storage state. Combined NMR and X-CT data reveal the following specific controls for the three lithotypes: ① Pore size determines fluid mobility patterns. NMR measurements show that coarse sandstone macropores and micro-fractures account for 48% and fully movable porosity for 68%. The large pore space provides ample room for fluid migration, allowing water molecules to form "preferential channel flow" along continuous conduits, yielding a permeability of 38.77 mD. Fine sandstone exhibits 19% micropores and 27% small pores with 64% immovable fraction; fluids are mostly adsorbed on pore surfaces or trapped in isolated spaces, migrating only through "slow diffusion under viscous control", giving a permeability of merely 0.87 mD. Medium sandstone contains 34% mesopores and 28% micropores, displaying a mixed "preferential channel flow plus local diffusion" regime with 1.46 mD permeability, confirming that larger pores correlate with higher movable-fluid fraction and greater flow efficiency. ② Pore connectivity governs seepage pathways. 3-D CT reconstructions indicate coarse sandstone has 16.19% connected porosity and 98.54% connectivity, forming a "densely interconnected mesh" with a mean throat radius of 32.63 µm and tortuosity of 1.63, offering straight flow paths. Fine sandstone shows 14.36% connected porosity and 97.82% connectivity, with pores mostly "isolated dots or short connections", mean throat radius 28.75 µm, and tortuosity 1.69, forcing fluid to detour and markedly increasing resistance. Average coordination numbers further verify this trend, being 8.05, 7.83, and 7.65 for coarse, medium, and fine sandstones, respectively; higher coordination yields more effective flow channels and stronger seepage capacity. ③ Fractal characteristics influence flow heterogeneity. NMR-derived fractal dimensions of connected pores are 2.955 for fine sandstone, 2.943 for medium sandstone, and 2.847 for coarse sandstone. The higher fractal dimension reflects stronger structural heterogeneity, meaning greater contrasts in pore size and more uneven spatial distribution, causing more frequent "sudden changes in flow resistance" during fluid migration and consequently the poorest seepage performance.

### 5.2. Numerical simulation

In sandstone, a typical porous medium, the fluid flow path is tortuous due to the complexity of the pore structure, and the flow velocity is low with significant viscous effects. These characteristics are consistent with the application conditions of the peristaltic flow model. Therefore, the peristaltic flow model was chosen in this study to simulate the seepage mechanism in the micro-pore structure of sandstone.

#### 5.2.1. Model description.
The Stokes equations can accurately describe the flow dominated by viscosity at low Reynolds numbers. When simulating single-phase peristaltic flow in COMSOL Multiphysics, the Stokes equations are adopted. These equations are a simplification of the Navier-Stokes equations (N – S equations) without considering

inertial forces. Meanwhile, it is assumed that the fluid is incompressible and satisfies the continuity equation. The specific set of equations is as follows:

$$\begin{cases} -\nabla p + \mu \nabla^2 \boldsymbol{u} = 0 \\ \nabla \cdot \boldsymbol{u} = 0 \end{cases} \tag{13}$$

In the equations, $\nabla p$ represents the pressure gradient, $\mu$ denotes the fluid viscosity, $\nabla^2$ is the Laplacian operator, and $u$ signifies the flow velocity.

Based on the 3D pore structure characteristic identification and seepage simulation analysis process in Section 4.1 (Fig 14), for the three types of sandstone samples, the pore structure volume size was determined to be 700μm × 700μm × 700μm in Step 1; the connected pore model was obtained in Step ③; and single – phase peristaltic flow simulation in the Z – direction was performed using the Stokes equations in the peristaltic flow module of Comsol Multiphysics in Step ⑤. This included the boundary condition parameter settings (Fig 14⑤c) as shown in Table 7 [43]. The pressure difference between the inlet and outlet is the main driving force for fluid flow in the sandstone pores. The greater the pressure difference, the stronger the driving force for fluid flow, and the higher the flow velocity. Conversely, the smaller the pressure difference, the lower the flow velocity. In this study, the default inlet and outlet pressures of the software were used.

**5.2.2. Simulation results and analysis.** In COMSOL, the velocity field and streamline diagram are two important visualization tools for displaying the characteristics of fluid flow. The velocity field is directly calculated and represents the velocity vectors of the fluid at each location, which can intuitively show the local flow characteristics of the fluid. It is suitable for scenarios where a precise understanding of the velocity magnitude and direction of the fluid at each location is required. On the other hand, the streamline diagram represents the direction and path of fluid flow through streamlines, which are curves tangent to the velocity field and indicate the trajectories of fluid particles. It can clearly show the overall flow path and direction of the fluid. Therefore, using both of these tools simultaneously can provide a more comprehensive understanding of the fluid flow characteristics, which is beneficial for engineering design and analysis.

The velocity field contour maps and streamline diagrams of the three types of sandstone samples were obtained from Step ⑤ of the simulation process (Fig 14). The velocity fields of the three types of sandstone are shown in Figs 16–18. It can be seen that all three types of sandstone are dominated by low-velocity flow. To further study the trajectory of fluid motion, streamlined diagrams of the three types of sandstone were drawn, as shown in Figs 19–21. It can be seen from Figs 19–21 that the fluid velocity is higher in the narrow parts of the pores and throats. The reason is that when the pores and throats are narrow, the fluid resistance increases, the fluid velocity distribution changes, the inertial effect becomes significant, and the pressure gradient changes. These factors together lead to an increase in fluid velocity in the narrow channels.

Further statistical analysis was conducted on the velocity distribution in the streamline diagrams, as shown in Fig 22. It can be seen from Fig 22 that the velocity field distribution of the three types of sandstone samples is dominated by low – velocity (less than 0.08 m/s) with a high frequency. As the velocity increases (greater than 0.08 m/s), the frequency of velocity distribution decreases rapidly. In the velocity range greater than 0.53 m/s, the frequency of velocity distribution for all three types of sandstone is less than 6%. This indicates that in the micro-pore seepage of sandstone, the inertial force of the fluid is much smaller than the viscous force, with the latter playing a dominant role. Low-velocity seepage is the core

**Table 7. Boundary condition parameters.**

| Fluid density $\rho$/kg·m$^{-3}$ | Kinematic viscosity $\mu$/Pa·s | Inlet water pressure $P_{in}$/Pa | Outlet water pressure $P_{out}$/Pa |
|---|---|---|---|
| 1000 | 0.001 | 130000 | 100000 |

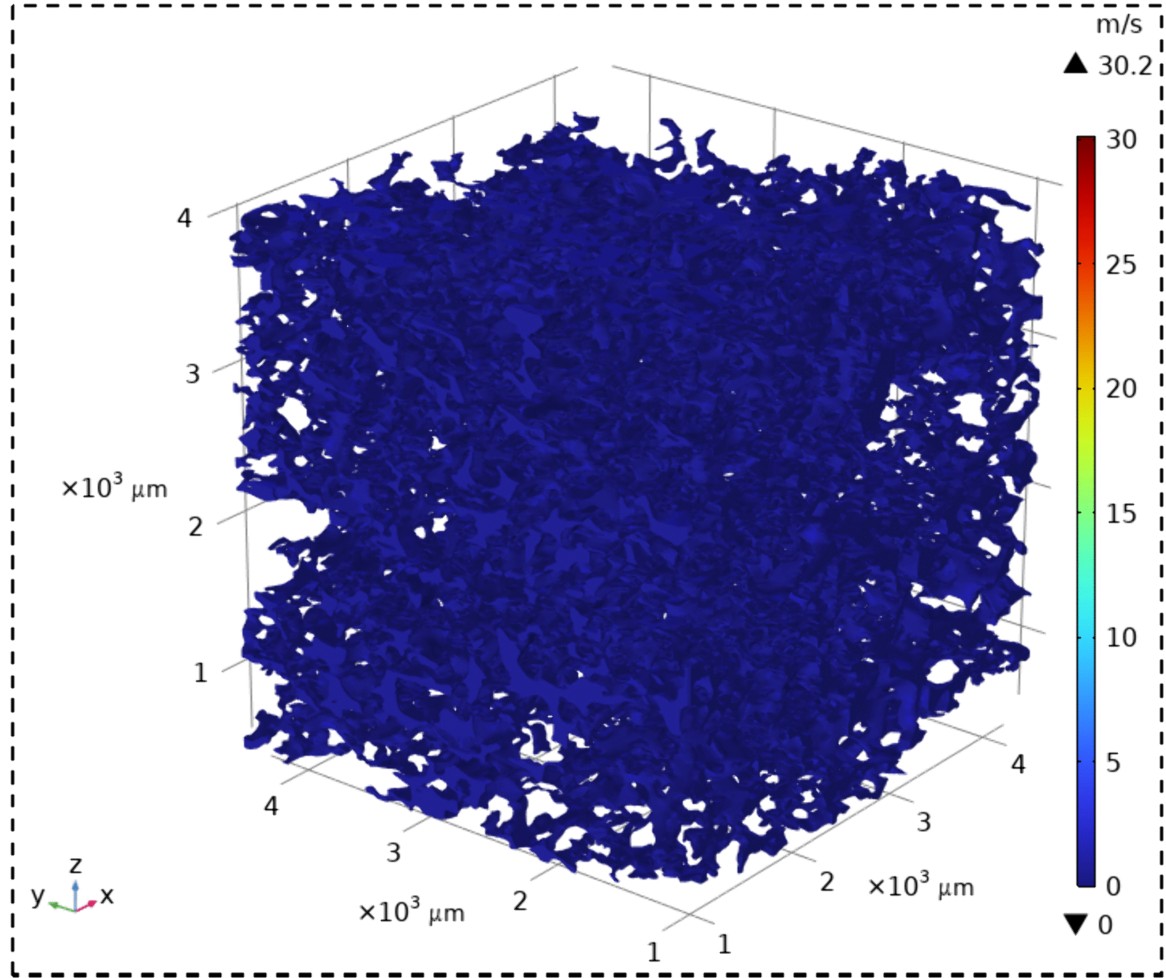

**Fig 16. Velocity field contour map of coarse sandstone.**

process. This also explains why the velocity field contour maps (Figs 16–18) and streamline diagrams (Fig 19 to Fig 21) do not show clear effects. The reason is that the higher velocities shown in the color scale of the velocity field of the three types of sandstone occur very rarely and are difficult to distinguish with the naked eye. For different sandstone samples, it can be seen from Fig 22 that in the velocity range less than 0.08 m/s, the frequency distribution from high to low is XSY (fine-grained sandstone) (97.54%), ZSY (medium-grained sandstone) (90.07%), and CSY (coarse-grained sandstone) (80.14%). In the other ranges greater than 0.08 m/s, the distribution frequency is highest for coarse-grained, followed by medium-grained, and then fine-grained sandstone. It is believed that this phenomenon is closely related to the pore structure and permeability of sandstone. Fine-grained sandstone has smaller grains, lower permeability, and higher tortuosity, which cause the fluid to experience more frictional resistance when flowing through it. In other words, the viscous effect is more pronounced, resulting in a predominance of low-velocity flow. Medium – and coarse-grained sandstones have larger grains and higher permeability, but lower tortuosity, which means that the fluid encounters less resistance when flowing through them, leading to higher velocities. Therefore, in the low-velocity frequency distribution, fine-grained sandstone has the highest proportion, followed by medium – and coarse-grained sandstone. However, the situation is reversed in the high-velocity frequency distribution. Coarse-grained sandstone has the highest proportion of high-velocity flow due to

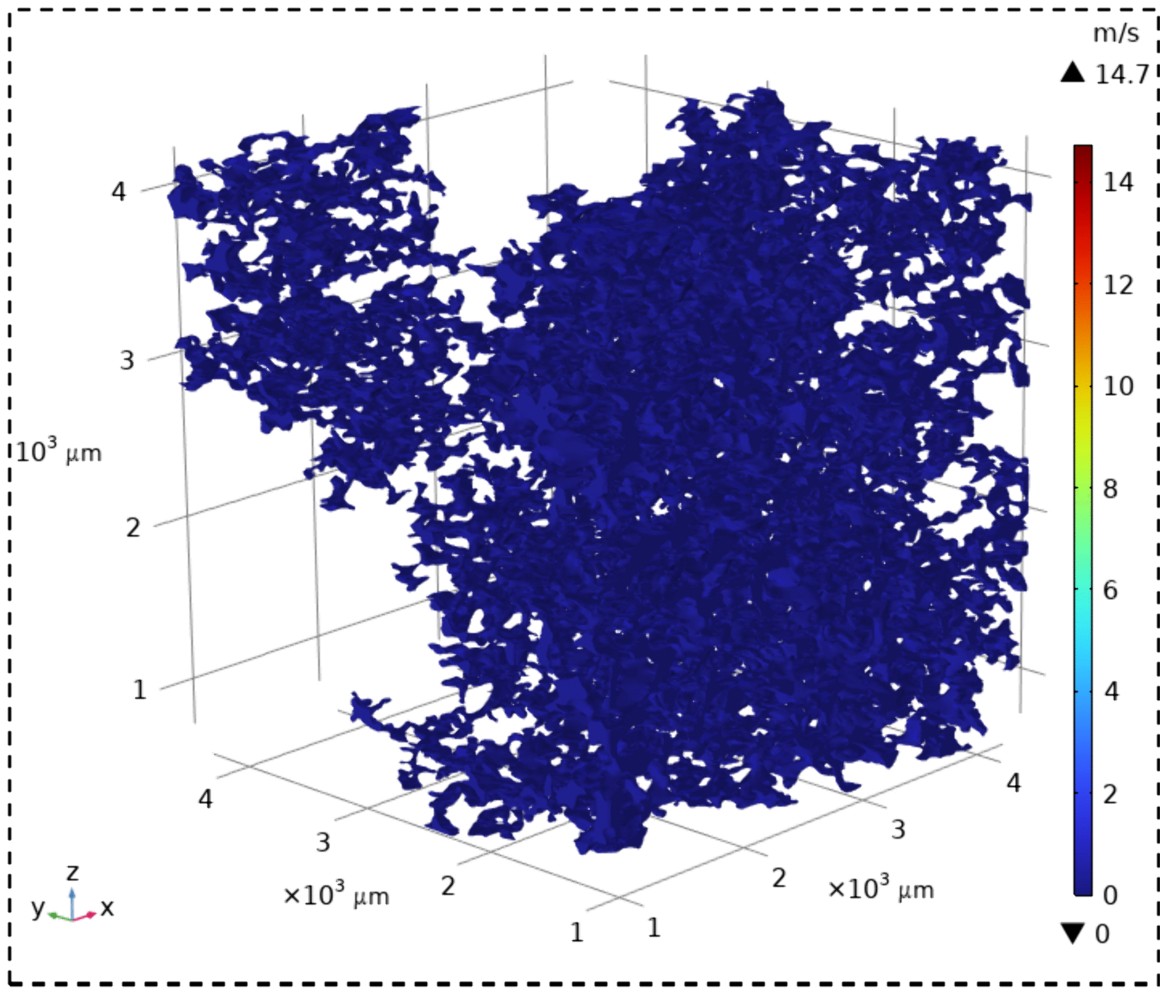

**Fig 17. Velocity field contour map of medium sandstone.**

its higher permeability and lower tortuosity, which allow fluid to flow at higher speeds. Medium-grained sandstone is next, while fine-grained sandstone has the lowest proportion of high-velocity flow because of its lowest permeability and highest tortuosity.

**5.2.3. Significance of micro-pore structure and seepage mechanism investigation.** By analyzing the micro-pore structure and seepage mechanism of different grain-size sandstones in the Luohe Formation of the Gaojiapu coal mine, Huanglong Jurassic coalfield, Ordos Basin, this study clarifies the intrinsic relationship between sandstone pore characteristics and flow capacity, providing key theoretical support and technical guidance for surface pre-grouting projects aimed at blocking and reducing roof water in coal mines. In practice, the rule that "coarse and medium sandstones, with their strong pore connectivity and well-developed throats, constitute the main migration pathways for roof water, whereas fine sandstones contain mostly isolated or poorly connected pores and contribute little to flow" should be used to focus grouting on coarse and medium sandstones while adopting simplified treatment for fine sandstones to avoid cost waste. For material and technique selection, high-fill composite slurries should be chosen for the macro-pore–fracture system of coarse and medium sandstones, and low-viscosity, high-penetration slurries for the micro-pores of

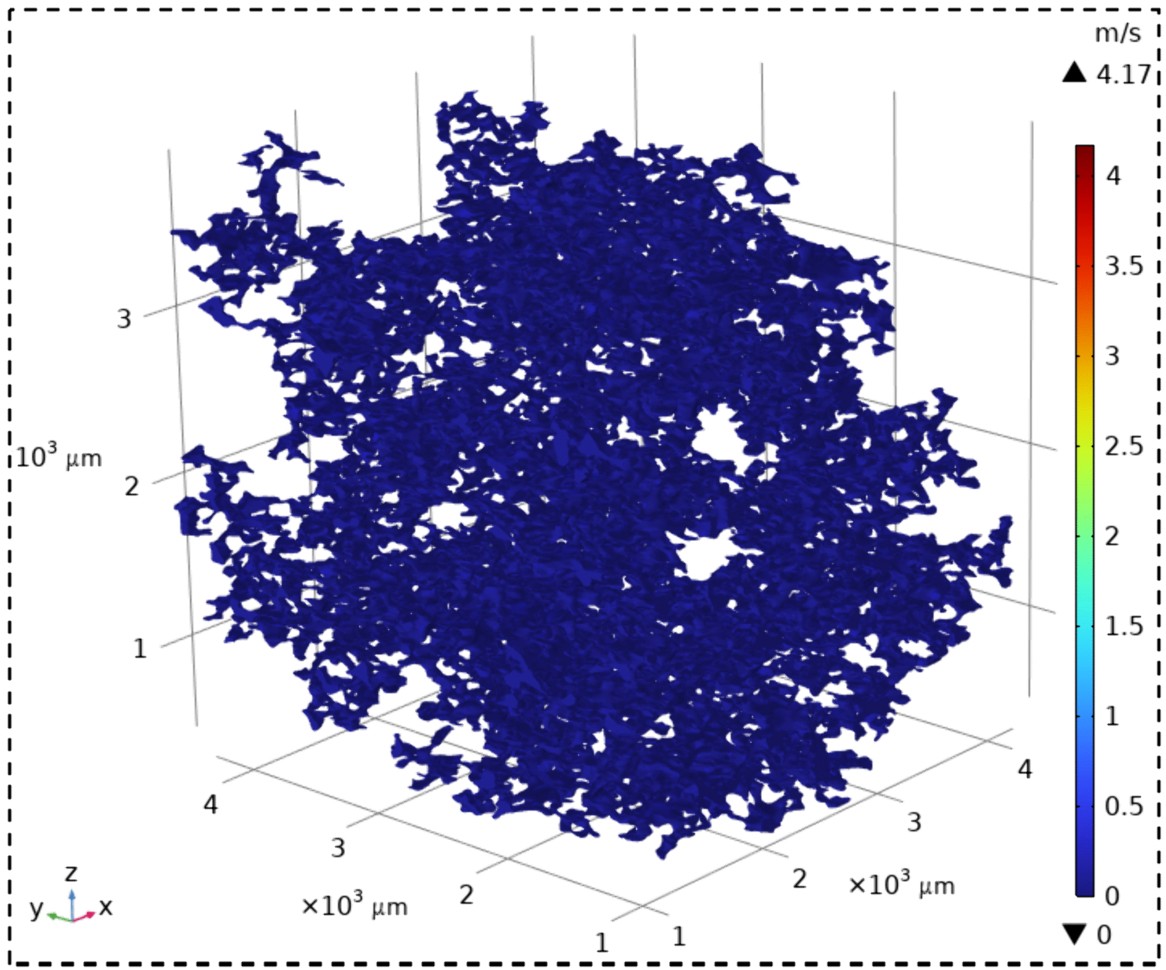

**Fig 18. Velocity field contour map of fine sandstone.**

fine sandstones; a staged pressure regime of "low-pressure diffusion plus medium-pressure consolidation" can balance spread radius and filling density. Meanwhile, borehole layout and construction monitoring should be optimized according to the different seepage responses: in coarse and medium sandstone zones, flow-rate and pressure sensors should be prioritized for early warning of water-inrush risk, whereas in fine sandstone zones, water-level tracking should be emphasized to judge plugging effectiveness. Integrating these findings into engineering design enables a scientifically based grouting scheme that ensures safe coal extraction while reducing both project cost and management blindness.

## 6. Conclusions

This paper investigates the pore structure and seepage characteristics of sandstone based on NMR and CT technologies. The conclusions are as follows:

(1) NMR Pore Parameters: Coarse-grained sandstone has a cumulative porosity of 14.36%, bound fluid porosity of 4.51%, movable fluid porosity of 9.85%, $T_{2cutoff1}$ of 1.70 ms, $T_{2cutoff2}$ of 16.83 ms, and permeability of 38.77 mD. Medium-grained sandstone has a cumulative porosity of 17.82%, bound fluid porosity of 11.62%, movable fluid

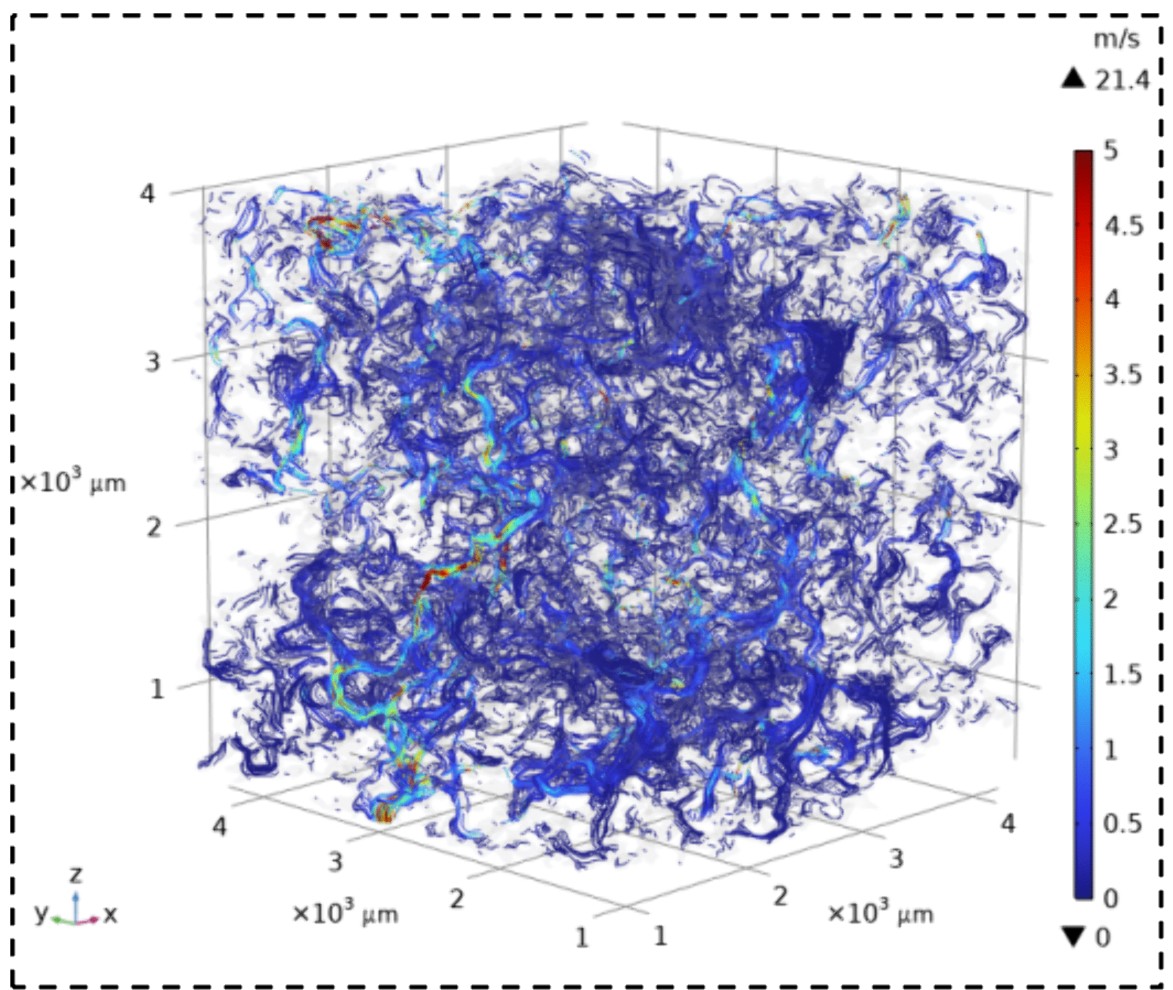

**Fig 19. Streamline map of seepage velocity in coarse sandstone.**

porosity of 6.20%, $T_{2cutoff1}$ of 9.66 ms, $T_{2cutoff2}$ of 20.73 ms, and permeability of 1.46 mD. Fine-grained sandstone has a cumulative porosity of 16.09%, bound fluid porosity of 10.73%, movable fluid porosity of 5.36%, $T_{2cutoff1}$ of 19.34 ms, $T_{2cutoff2}$ of 31.44 ms, and permeability of 0.87 mD. Coarse-grained sandstone has a large reservoir capacity and good fluid mobility, while fine-grained sandstone has the best reservoir capacity but the worst fluid mobility.

(2) NMR Pore Distribution: The proportions of micropores, small pores, medium pores, large pores, and micro-fractures in coarse-grained sandstone are 7%, 18%, 48%, and 27%, respectively. Those in medium-grained sandstone are 28%, 26%, 34%, and 12%, respectively. Those in fine-grained sandstone are 19%, 27%, 46%, and 8%, respectively. The proportion of fully movable pores is the highest in coarse-grained sandstone, reaching 68%, while the proportion of fully bound pores is the highest in fine-grained sandstone, reaching 64%.

(3) NMR Fractal Dimension: The connected pore fractal dimension of coarse-grained sandstone is 2.847, and the isolated pore fractal dimension is 1.436. The connected pore fractal dimension of medium-grained sandstone is 2.943, and the isolated pore fractal dimension is 0.981. The connected pore fractal dimension of fine-grained sandstone is 2.955, and

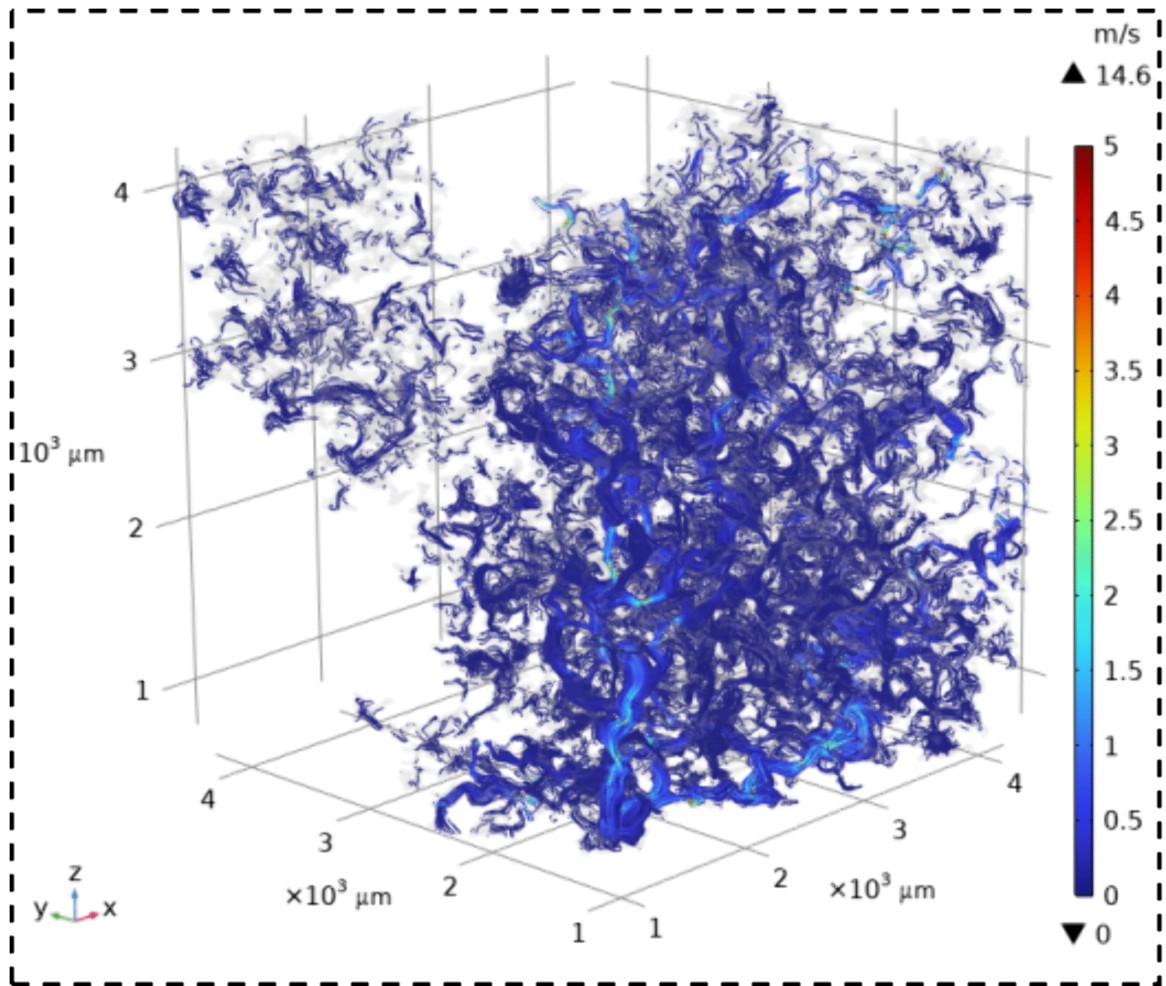

**Fig 20. Streamline map of seepage velocity in medium sandstone.**

the isolated pore fractal dimension is 1.033. This indicates that coarse-grained sandstone is the most conducive to fluid flow, while fine-grained sandstone is the most conducive to fluid storage.

(4) 3D Reconstructed Pore Parameters: Coarse-grained sandstone has a total porosity of 16.43%, connected porosity of 16.19%, connectivity of 98.54%, and total pore fractal dimension of 1.53. Medium-grained sandstone has a total porosity of 15.51%, connected porosity of 15.21%, connectivity of 98.07%, and total pore fractal dimension of 1.51. Fine-grained sandstone has a total porosity of 14.68%, connected porosity of 14.36%, connectivity of 97.82%, and total pore fractal dimension of 1.50. This indicates that coarse-grained sandstone has the highest degree of pore development and spatial complexity, while fine-grained sandstone has the lowest.

(5) 3D Reconstructed Pore Distribution: Coarse-grained sandstone has a mean pore diameter of 17.50 μm and a mean pore-throat radius of 32.63 μm, both of which are larger than those of medium-grained sandstone (mean pore diameter of 17.16 μm and mean pore – throat radius of 29.11 μm) and fine-grained sandstone (mean pore diameter of 17.03

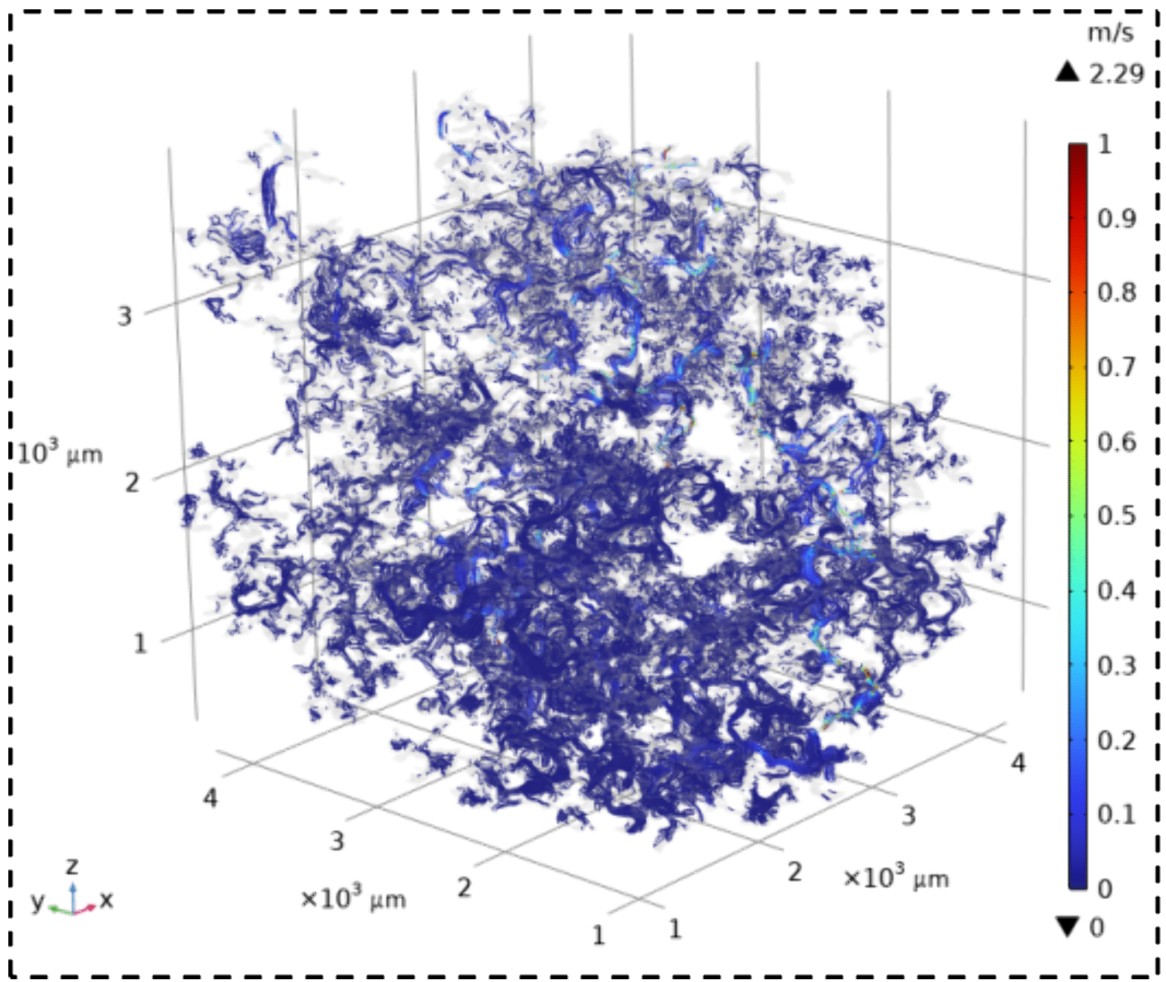

**Fig 21. Streamline map of seepage velocity in fine sandstone.**

µm and mean pore – throat radius of 28.75 µm). This indicates that coarse-grained sandstone has larger pore and pore-throat sizes, which are conducive to fluid flow, while fine-grained sandstone has relatively higher flow resistance.

(6) Seepage Simulation Results: Coarse-grained sandstone has the highest absolute permeability (10.88 D), while fine-grained sandstone has the lowest absolute permeability (8.40 D). In terms of tortuosity, coarse-grained sandstone has the smallest tortuosity (1.63), indicating the simplest fluid flow path. Fine-grained sandstone has the largest tortuosity (1.69), indicating the most complex flow path. In terms of coordination number, coarse-grained sandstone has the highest mean value of 8.05, indicating the most complex pore network connections. Fine-grained sandstone has the lowest mean value of 7.65, indicating the simplest network connections. The velocity field distribution of the three types of sandstone samples is dominated by low – velocity (less than 0.8 m/s) with a high frequency. In the velocity range greater than 0.53 m/s, the frequency of velocity distribution for all three types of sandstone is less than 6%. In the velocity range of fewer than 0.08 m/s, the frequency distribution from high to low is fine-grained sandstone (97.54%), medium-grained sandstone (90.07%), and coarse-grained sandstone (80.14%). In the other ranges greater than 0.08 m/s, the distribution frequency is highest for coarse-grained, followed by medium-grained, and then fine-grained sandstone.

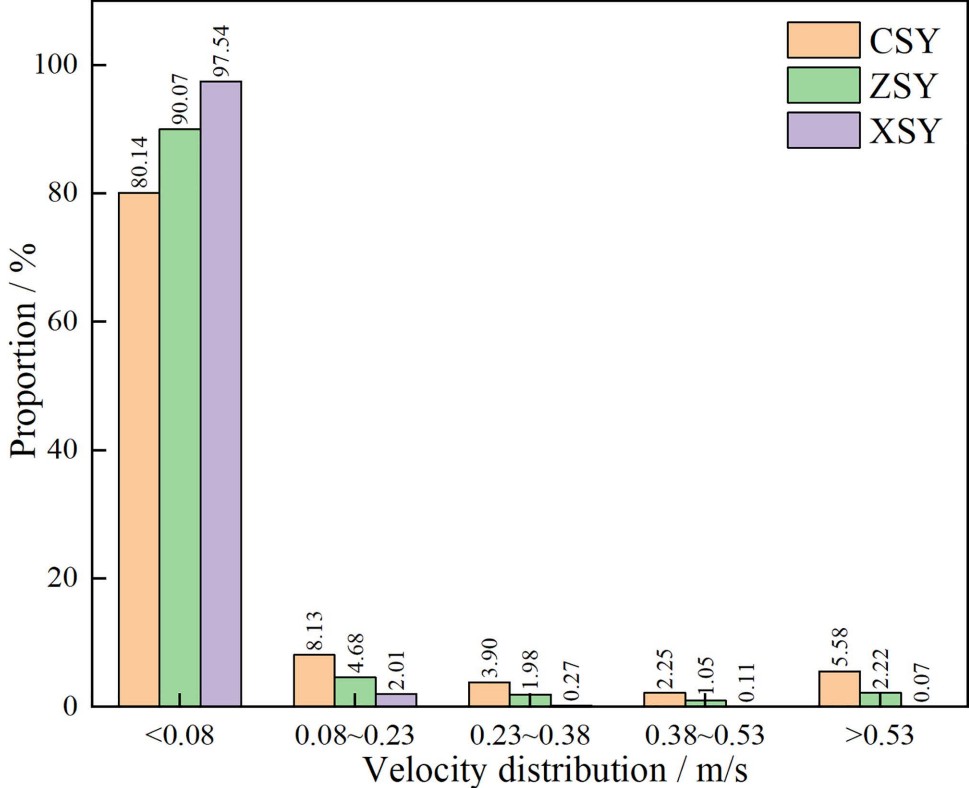

**Fig 22. Histogram of velocity distribution frequency for three types of sandstone.**

## Supporting information

**S1 Data. All raw data underlying the figures and tables are provided in the minimal data set file.**
(RAR)

## Author contributions

**Conceptualization:** Kaide Liu.

**Data curation:** Songxin Zhao, Kaide Liu.

**Formal analysis:** Songxin Zhao.

**Funding acquisition:** Kaide Liu.

**Investigation:** Songxin Zhao.

**Methodology:** Songxin Zhao, Kaide Liu.

**Project administration:** Kaide Liu.

**Resources:** Kaide Liu.

**Software:** Songxin Zhao, Kaide Liu, Qiyu Wang, Yu Xia, Xinping Wang.

**Supervision:** Kaide Liu, Chaowei Sun, Wenping Yue.

**Validation:** Songxin Zhao.

**Visualization:** Songxin Zhao, Kaide Liu.

**Writing – original draft:** Songxin Zhao.

**Writing – review & editing:** Kaide Liu.

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
