## [Decision Letter · Decision Letter 0]

22 Sep 2025

Dear Dr. Liu, 

Thank you for submitting your manuscript to PLOS ONE. After careful consideration, we feel that it has merit but does not fully meet PLOS ONE’s publication criteria as it currently stands. Therefore, we invite you to submit a revised version of the manuscript that addresses the points raised during the review process.

We look forward to receiving your revised manuscript.

Kind regards,

Hu Li

Academic Editor

PLOS ONE

Journal Requirements:

3. We note that this submission includes NMR spectroscopy data. We would recommend that you include the following information in your methods section or as Supporting Information files:

1) The make/source of the NMR instrument used in your study, as well as the magnetic field strength. For each individual experiment, please also list: the nucleus being measured; the sample concentration; the solvent in which the sample is dissolved and if solvent signal suppression was used; the reference standard and the temperature.

2) A list of the chemical shifts for all compounds characterised by NMR spectroscopy, specifying, where relevant: the chemical shift (δ), the multiplicity and the coupling constants (in Hz), for the appropriate nuclei used for assignment.

3)The full integrated NMR spectrum, clearly labelled with the compound name and chemical structure.

We also strongly encourage authors to provide primary NMR data files, in particular for new compounds which have not been characterised in the existing literature. Authors should provide the acquisition data, FID files and processing parameters for each experiment, clearly labelled with the compound name and identifier, as well as a structure file for each provided dataset. 

See our list of recommended repositories here: https://journals.plos.org/plosone/s/recommended-repositories

“This work is supported by The National Nature Science Foundation of China (No. 52104222, No. 51909224), the Natural Science Foundation Research Project of Shaanxi Province (2021JLM-48, 2025JCBMS-511, 2019JM-182), and the Special Fund for High-level Talents of Xijing University (XJ24B12, XJ18T04).”

5. We note that your Data Availability Statement is currently as follows: 

“All relevant data are within the manuscript and its Supporting Information files.”

6. We note that Figure 8 in your submission contain copyrighted images. All PLOS content is published under the Creative Commons Attribution License (CC BY 4.0), which means that the manuscript, images, and Supporting Information files will be freely available online, and any third party is permitted to access, download, copy, distribute, and use these materials in any way, even commercially, with proper attribution. For more information, see our copyright guidelines: http://journals.plos.org/plosone/s/licenses-and-copyright.

1) You may seek permission from the original copyright holder of Figure 8 to publish the content specifically under the CC BY 4.0 license. 

2) If you are unable to obtain permission from the original copyright holder to publish these figures under the CC BY 4.0 license or if the copyright holder’s requirements are incompatible with the CC BY 4.0 license, please either i) remove the figure or ii) supply a replacement figure that complies with the CC BY 4.0 license. Please check copyright information on all replacement figures and update the figure caption with source information. If applicable, please specify in the figure caption text when a figure is similar but not identical to the original image and is therefore for illustrative purposes only.

Reviewers' comments:

Reviewer's Responses to Questions

**Comments to the Author**

1. Is the manuscript technically sound, and do the data support the conclusions?

Reviewer #1: Yes

Reviewer #2: Yes

2. Has the statistical analysis been performed appropriately and rigorously?

Reviewer #1: Yes

Reviewer #2: Yes

3. Have the authors made all data underlying the findings in their manuscript fully available?

Reviewer #1: Yes

Reviewer #2: Yes

4. Is the manuscript presented in an intelligible fashion and written in standard English?

Reviewer #1: Yes

Reviewer #2: Yes

Reviewer #1: This manuscript is based on a combination of multiple methods to study the Pore Structure and Permeability Simulation of sandstone. It can be seen that the authors have carried out a certain degree of work and conducted detailed experimental analysis. The following suggestions should help the authors better revise the manuscript.

1)The figures displaying experimental samples lack a scale bar.

2�Regarding the images of instruments and experimental processes, it is recommended to further integrate them through logical link diagrams or flowcharts.

3�This manuscript provides a detailed analysis of the experiment, but what kind of patterns can be obtained from the data obtained? New acquaintance? How to guide problems in production practice? The discussion on these aspects is relatively lacking, it is suggested to further supplement.

4�It is suggested that the authors further optimize the language and supplement the references from the past three years.

Reviewer #2: 1. This study employs NMR and CT techniques to quantitatively compare the porosity, pore-size distribution, pore type, connectivity, and permeability of different sandstones. Digital cores reconstructed from CT slices are then used to perform pore-scale flow simulations in Comsol, providing a theoretical basis for preventing roof-water hazards in coal seams.

2. Title ambiguity: change “Study on Characterization of Sandstone Pore Structure and Permeability Simulation Based on NMR and CT Technology” to “Study on Characterization of Sandstone Pore Structure and Seepage Mechanism Based on NMR and CT Technology”.

3. The abstract is verbose and imprecise; condense it to highlight purpose, methods, conclusions, and engineering significance while tightening the logic.

4. Lines 58~60 of the Introduction (“Existing grouting has sealed most pores and fractures; however, tiny pores still leak continuously, so sandstone-roof water hazards remain serious”) require literature support.

5. The Introduction should summarize the limitations of previous work on pore structure and seepage mechanism, with citations, to strengthen the motivation for this study.

6. Add a brief description of the geological setting of the tested samples at the end of the Introduction.

7. Sections 3 and 4.1~4.2 analyze porosity, permeability, fractal dimension, tortuosity, etc., but the theoretical link between pore structure and seepage mechanism is unclear; insert a new discussion section before the flow simulations.

8. Formatting errors (lines 359, 377~387, 423, 429, 486), such as misaligned equations, must be corrected throughout.

**Do you want your identity to be public for this peer review?** For information about this choice, including consent withdrawal, please see our Privacy Policy

Reviewer #1: No

Reviewer #2: No

---

## [Author Response · Author response to Decision Letter 1]

16 Oct 2025

Dear Editor-in-Chief:

After carefully and meticulously studying the reviewers' comments, the author has thoroughly re-examined the manuscript and made the following revisions and explanations in accordance with the reviewers' opinions and requirements. Additionally, through further analysis and consideration, the author has also revised the parts that the reviewers did not mention but the author believes should be clarified more explicitly. These revisions are highlighted in red for your verification.

Academic Editor 1

Comment 1. In your Methods section, please provide additional information regarding the permits you obtained for the work. Please ensure you have included the full name of the authority that approved the field site access and, if no permits were required, a brief statement explaining why.

The author agrees with and accepts the expert's suggestions. We clarify as follows: The specimens used in this study were purchased commercially; no additional permissions are required. The purchase invoice is provided in Figure 1.

Figure. 1 Purchase invoice

Comment 2. Thank you for stating the following financial disclosure:

“This work is supported by The National Nature Science Foundation of China (No. 52104222, No. 51909224), the Natural Science Foundation Research Project of Shaanxi Province (2021JLM-48, 2025JCBMS-511, 2019JM-182), and the Special Fund for High-level Talents of Xijing University (XJ24B12, XJ18T04).”

The author agrees with and accepts the expert's suggestions.We hereby clarify that all funders played specific roles in this study. Their contributions are detailed below and have been included in the cover letter:Conceptualization: Kaide Liu, Data curation: Kaide Liu, Songxin Zhao, Formal analysis: Kaide Liu, Songxin Zhao, Funding acquisition: Kaide Liu, Investigation:,Songxin Zhao, Methodology: Kaide Liu, Songxin Zhao, Project administration: Kaide Liu, Resources: Kaide Liu, Software: Kaide Liu, Songxin Zhao, Yu Xia, Qiyu Wang, Xinping Wang, Supervision: Kaide Liu, Wenping Yue, Chaowei Sun, Validation: Songxin Zhao ,Visualization: Kaide Liu, Songxin Zhao ,Writing-original draft: Songxin Zhao ,Writing - review & editing: Kaide Liu.

Comment 3. Thank you for updating your data availability statement. You note that your data are available within the Supporting Information files, but no such files have been included with your submission. At this time we ask that you please upload your minimal data set as a Supporting Information file, or to a public repository such as Figshare or Dryad.

Please also ensure that when you upload your file you include separate captions for your supplementary files at the end of your manuscript.

As soon as you confirm the location of the data underlying your findings, we will be able to proceed with the review of your submission.

Thank you for the editor’s comment. The “minimal data set” has now been uploaded as part of the Supporting Information file. A separate heading for the supplementary file has also been added at the end of the manuscript.

Comment 4. We note that your manuscript is not formatted using one of PLOS ONE’s accepted file types. Please reattach your manuscript as one of the following file types: .doc, .docx, .rtf, or .tex (accompanied by a .pdf).

If your submission was prepared in LaTex, please submit your manuscript file in PDF format and attach your .tex file as “other.”

Thank you for the editor’s comment. The file has been converted to .docx format.

Academic Editor 2

Comment 1. Please ensure that your manuscript meets PLOS ONE's style requirements, including those for file naming. The PLOS ONE style templates can be found at https://journals.plos.org/plosone/s/file?id=wjVg/PLOSOne_formatting_sample_main_body.pdf and https://journals.plos.org/plosone/s/file?id=ba62/PLOSOne_formatting_sample_title_authors_affiliations.pdf.

The author agrees with and accepts the expert's suggestions, and has made the necessary revisions.

Comment 2. In your Methods section, please provide additional information regarding the permits you obtained for the work. Please ensure you have included the full name of the authority that approved the field site access and, if no permits were required, a brief statement explaining why.

Answer in line with Editor 1’ s Comment 1.

Comment 3. We note that this submission includes NMR spectroscopy data. We would recommend that you include the following information in your methods section or as Supporting Information files:

1) The make/source of the NMR instrument used in your study, as well as the magnetic field strength. For each individual experiment, please also list: the nucleus being measured; the sample concentration; the solvent in which the sample is dissolved and if solvent signal suppression was used; the reference standard and the temperature.

2) A list of the chemical shifts for all compounds characterised by NMR spectroscopy, specifying, where relevant: the chemical shift (δ), the multiplicity and the coupling constants (in Hz), for the appropriate nuclei used for assignment.

3)The full integrated NMR spectrum, clearly labelled with the compound name and chemical structure.

We also strongly encourage authors to provide primary NMR data files, in particular for new compounds which have not been characterised in the existing literature. Authors should provide the acquisition data, FID files and processing parameters for each experiment, clearly labelled with the compound name and identifier, as well as a structure file for each provided dataset.

See our list of recommended repositories here: https://journals.plos.org/plosone/s/recommended-repositories

The author agrees with and accepts the expert's suggestions. Low-field nuclear magnetic resonance (LF-NMR) was employed in this study to characterize the water distribution and porosity of the sandstone pore structure; no elucidation of organic molecular structures was involved. Consequently, conventional high-resolution NMR parameters—such as chemical shifts, coupling constants, solvents, or reference standards—are not required. All raw data have been provided in the Supporting Information.

Thank you for stating the following financial disclosure:

Comment 4. “This work is supported by The National Nature Science Foundation of China (No. 52104222, No. 51909224), the Natural Science Foundation Research Project of Shaanxi Province (2021JLM-48, 2025JCBMS-511, 2019JM-182), and the Special Fund for High-level Talents of Xijing University (XJ24B12, XJ18T04).”

Answer in line with Editor 1’ s Comment 2.

Comment 5. We note that your Data Availability Statement is currently as follows:

“All relevant data are within the manuscript and its Supporting Information files.”

The author agrees with and accepts the expert's suggestions. The minimal data set has been uploaded.

Comment 6. We note that Figure 8 in your submission contain copyrighted images. All PLOS content is published under the Creative Commons Attribution License (CC BY 4.0), which means that the manuscript, images, and Supporting Information files will be freely available online, and any third party is permitted to access, download, copy, distribute, and use these materials in any way, even commercially, with proper attribution. For more information, see our copyright guidelines: http://journals.plos.org/plosone/s/licenses-and-copyright.

1) You may seek permission from the original copyright holder of Figure 8 to publish the content specifically under the CC BY 4.0 license.

2)If you are unable to obtain permission from the original copyright holder to publish these figures under the CC BY 4.0 license or if the copyright holder’s requirements are incompatible with the CC BY 4.0 license, please either i) remove the figure or ii) supply a replacement figure that complies with the CC BY 4.0 license. Please check copyright information on all replacement figures and update the figure caption with source information. If applicable, please specify in the figure caption text when a figure is similar but not identical to the original image and is therefore for illustrative purposes only.

The author agrees with and accepts the expert's suggestions. The figure has been removed.

Comment 7. If the reviewer comments include a recommendation to cite specific previously published works, please review and evaluate these publications to determine whether they are relevant and should be cited. There is no requirement to cite these works unless the editor has indicated otherwise.

The author agrees with and accepts the expert's suggestions. The reviewers did not request citation of any specific references.

Comment 8. Please review your reference list to ensure that it is complete and correct. If you have cited papers that have been retracted, please include the rationale for doing so in the manuscript text, or remove these references and replace them with relevant current references. Any changes to the reference list should be mentioned in the rebuttal letter that accompanies your revised manuscript. If you need to cite a retracted article, indicate the article’s retracted status in the References list and also include a citation and full reference for the retraction notice.

The author agrees with and accepts the expert's suggestions. All references are complete and accurate.

Reviewer 1

Comment 1. The figures displaying experimental samples lack a scale bar.

The author agrees with and accepts the expert's suggestions. The specific modifications in the text are as follows:

2. Experimental methods

2.1. NMR experiments

The detailed NMR experimental steps are shown in Fig 1, specifically as follows: ① Sample origin: Core samples of Luhe Formation sandstone were taken from the west wing of Panel 1 in Gaojiapu Coal Mine, Binchang Mining Area. Among them, coarse and medium sandstones were from the middle section of the Luhe Formation, while fine sandstone was from the lower section, with colors mainly light purplish red and light brownish red. ② Sample preparation: Using a Z3032X8 radial drilling machine and an SPQJ-300 slicing machine, the cores were processed into cylindrical samples with a diameter of 25 mm and a height of 50 mm. The coarse, medium, and fine sandstones were labeled as CSY, ZSY, and XSY, respectively, arranged from left to right as CSY, ZSY, and XSY. ③ Vacuum saturation: The samples were placed in a ZYB-Ⅱ vacuum pressure saturation device, first vacuumed for 8 hours, then water was injected and pressurized to 8 MPa and maintained for more than 24 hours to ensure full saturation of pores. ④ NMR measurement: The samples were placed into a MacroMR12-150H-I low-field nuclear magnetic resonance instrument (Xijing University), and the equipment parameters were set as follows: 25 mm coil, CPMG sequence, O1=12773.43 Hz, P1=8 μs, P2=14.48 μs, SW=250KHz, PRG=2, TW=1500 ms, NS=8, TE=0.06 ms, NECH echo number=18000, to obtain the T2 spectrum under saturated conditions. ⑤ Centrifugation treatment: The samples were placed in an H3-18K desktop high-speed centrifuge, set at 4000 r/min, and centrifuged for 15 minutes to remove movable water; then, they were tested again to obtain the T2 spectrum of bound fluid.

Fig 1. NMR experimental procedure

2.2 CT Experiments

The detailed CT experimental steps are shown in Fig 2 and are as follows: ① Sample origin: identical to those used in the NMR test. ② Sample preparation: cores were first rough-cut with an SPQJ-300 diamond saw, further trimmed with a metallographic cutter, and finally machined into 5 mm × 5 mm × 15 mm rectangular prisms using an LC-200XP automatic high-precision cutter (Nabo, Jiaxing). From left to right, the nine specimens comprised three fine-, three medium-, and three coarse-sandstone prisms; defect-free samples were selected for each lithotype and coded XSY, ZSY, and CSY, respectively. ③ Sample loadting: prisms were loaded into a Zeiss Xradia 510 Versa high-resolution 3D X-ray microscope (Guilin University of Technology) operated with (1) 3-D spatial resolution < 0.7 µm, (2) X-ray tube voltage 30–160 kV, maximum power 10 W, (3) 2k × 2k CCD camera, (4) interchangeable objective lenses (0.4X, 4X) manually selected according to resolution requirements, (5) detector travel range 290 mm, (6) propagation-based phase-contrast imaging, (7) maximum sample width 300 mm, and (8) maximum sample mass (including holder) 15 kg. ④ Test initiated: continuous transverse sections were scanned for each sandstone prism, yielding 1000 two-dimensional sequential images (TIFF format).

Fig 2. CT experimental procedure

(As marked in red in the text, see lines 111 to 155)

Comment 2. Regardin

---

## [Decision Letter · Decision Letter 1]

4 Nov 2025

Dear Dr. Liu,

Thank you for submitting your manuscript to PLOS ONE. After careful consideration, we feel that it has merit but does not fully meet PLOS ONE’s publication criteria as it currently stands. Therefore, we invite you to submit a revised version of the manuscript that addresses the points raised during the review process.

We look forward to receiving your revised manuscript.

Kind regards,

Hu Li

Academic Editor

PLOS ONE

Journal Requirements:

**Additional Editor Comments:**

Dear authors,

Reviewer 2 has accepted the paper, whereas Reviewer 1 has not responded. Given that Reviewer 1’s comments were relatively minor and considering my own evaluation, the following revisions are required before the paper can be accepted:

(1) There is one point raised by Reviewer 1 that was not adequately addressed in the authors’ response. Please review it again:“This manuscript provides a detailed analysis of the experiment, but what kind of patterns can be obtained from the data obtained? New acquaintance? How to guide problems in production practice? The discussion on these aspects is relatively lacking; it is suggested to further supplement.”

(2) The figures in the paper still do not fully meet standard formatting requirements. Please revise each one accordingly.

(3) The references should be standardized. In particular, it is essential to investigate and cite relevant papers published in the last three years.

Once these issues are resolved, the paper will be officially accepted without further review by the reviewers.

Reviewers' comments:

Reviewer's Responses to Questions

**Comments to the Author**

1. If the authors have adequately addressed your comments raised in a previous round of review and you feel that this manuscript is now acceptable for publication, you may indicate that here to bypass the “Comments to the Author” section, enter your conflict of interest statement in the “Confidential to Editor” section, and submit your "Accept" recommendation.

Reviewer #2: All comments have been addressed

2. Is the manuscript technically sound, and do the data support the conclusions?

Reviewer #2: Yes

3. Has the statistical analysis been performed appropriately and rigorously?

Reviewer #2: Yes

4. Have the authors made all data underlying the findings in their manuscript fully available?

Reviewer #2: Yes

5. Is the manuscript presented in an intelligible fashion and written in standard English?

Reviewer #2: Yes

Reviewer #2: (No Response)

**Do you want your identity to be public for this peer review?** For information about this choice, including consent withdrawal, please see our Privacy Policy

Reviewer #2: No

---

## [Author Response · Author response to Decision Letter 2]

5 Nov 2025

Dear Editor-in-Chief:

After carefully and meticulously studying the reviewers' comments, the author has thoroughly re-examined the manuscript and made the following revisions and explanations in accordance with the reviewers' opinions and requirements. Additionally, through further analysis and consideration, the author has also revised the parts that the reviewers did not mention but the author believes should be clarified more explicitly. These revisions are highlighted in red for your verification.

Reviewer 1

Comment 1. There is one point raised by Reviewer 1 that was not adequately addressed in the authors’ response. Please review it again:“This manuscript provides a detailed analysis of the experiment, but what kind of patterns can be obtained from the data obtained? New acquaintance? How to guide problems in production practice? The discussion on these aspects is relatively lacking; it is suggested to further supplement.”

The author agrees with and accepts the expert's suggestions. The specific modifications in the text are as follows:

5.2.3 Significance of micro-pore structure and seepage mechanism investigation

By analyzing the micro-pore structure and seepage mechanism of different grain-size sandstones in the Luohe Formation of the Gaojiapu coal mine, Huanglong Jurassic coalfield, Ordos Basin, this study clarifies the intrinsic relationship between sandstone pore characteristics and flow capacity, providing key theoretical support and technical guidance for surface pre-grouting projects aimed at blocking and reducing roof water in coal mines. In practice, the rule that “coarse and medium sandstones, with their strong pore connectivity and well-developed throats, constitute the main migration pathways for roof water, whereas fine sandstones contain mostly isolated or poorly connected pores and contribute little to flow” should be used to focus grouting on coarse and medium sandstones while adopting simplified treatment for fine sandstones to avoid cost waste. For material and technique selection, high-fill composite slurries should be chosen for the macro-pore–fracture system of coarse and medium sandstones, and low-viscosity, high-penetration slurries for the micro-pores of fine sandstones; a staged pressure regime of “low-pressure diffusion plus medium-pressure consolidation” can balance spread radius and filling density. Meanwhile, borehole layout and construction monitoring should be optimized according to the different seepage responses: in coarse and medium sandstone zones, flow-rate and pressure sensors should be prioritized for early warning of water-inrush risk, whereas in fine sandstone zones, water-level tracking should be emphasized to judge plugging effectiveness. Integrating these findings into engineering design enables a scientifically based grouting scheme that ensures safe coal extraction while reducing both project cost and management blindness.

(As marked in red in the text, see lines 672 to 695)

Comment 2. The figures in the paper still do not fully meet standard formatting requirements. Please revise each one accordingly.

The author agrees with and accepts the expert's suggestions. All figures and tables have been reformatted to comply with the journal’s specifications.

Comment 3. The references should be standardized. In particular, it is essential to investigate and cite relevant papers published in the last three years.

The author agrees with and accepts the expert's suggestions. The references have been standardized as requested.

---

## [Editor Report · Decision Letter 2]

10 Nov 2025

Study on Characterization of Sandstone Pore Structure and Seepage Mechanism Based on NMR and CT Technology

PONE-D-25-37332R2

Dear Dr. Liu,

We’re pleased to inform you that your manuscript has been judged scientifically suitable for publication and will be formally accepted for publication once it meets all outstanding technical requirements.

Kind regards,

Hu Li

Academic Editor

PLOS ONE

Additional Editor Comments (optional):

The author revised the paper twice, and the quality of the paper was significantly improved. Therefore, the paper can be accepted for publication.
---

## [Editor Report · Acceptance letter]

PONE-D-25-37332R2

PLOS One

Dear Dr. Liu,

I'm pleased to inform you that your manuscript has been deemed suitable for publication in PLOS One. Congratulations! Your manuscript is now being handed over to our production team.

Kind regards,

on behalf of

Pro.Dr. Hu Li

Academic Editor

PLOS One